# Rvb1/Rvb2 proteins couple transcription and translation during glucose starvation

**Yang S Chen[1,2], Wanfu Hou[2], Sharon Tracy[2], Alex T Harvey[2], Vince Harjono[2], Fan Xu[1], James J Moresco[3], John R Yates III[3], Brian M Zid[2]\***

[1]Division of Biological Sciences, University of California, San Diego, San Diego, United States; [2]Department of Chemistry and Biochemistry, University of California San Diego, San Diego, United States; [3]Department of Chemical Physiology, The Scripps Research Institute, La Jolla, United States

**Abstract:** During times of unpredictable stress, organisms must adapt their gene expression to maximize survival. Along with changes in transcription, one conserved means of gene regulation during conditions that quickly repress translation is the formation of cytoplasmic phase-separated mRNP granules such as P-bodies and stress granules. Previously, we identified that distinct steps in gene expression can be coupled during glucose starvation as promoter sequences in the nucleus are able to direct the subcellular localization and translatability of mRNAs in the cytosol. Here, we report that Rvb1 and Rvb2, conserved ATPase proteins implicated as protein assembly chaperones and chromatin remodelers, were enriched at the promoters and mRNAs of genes involved in alternative glucose metabolism pathways that we previously found to be transcriptionally upregulated but translationally downregulated during glucose starvation in yeast. Engineered Rvb1/Rvb2-binding on mRNAs was sufficient to sequester mRNAs into mRNP granules and repress their translation. Additionally, this Rvb tethering to the mRNA drove further transcriptional upregulation of the target genes. Further, we found that depletion of Rvb2 caused decreased alternative glucose metabolism gene mRNA induction, but upregulation of protein synthesis during glucose starvation. Overall, our results point to Rvb1/Rvb2 coupling transcription, mRNA granular localization, and translatability of mRNAs during glucose starvation. This Rvb-mediated rapid gene regulation could potentially serve as an efficient recovery plan for cells after stress removal.

**\*For correspondence:**
zid@ucsd.edu

**Competing interest:** The authors declare that no competing interests exist.

## Editor's evaluation

This study provides convincing evidence that the Rvb1 and Rvb2 proteins preferentially associate with the promoters of a class of genes that exhibit increased mRNA expression but translational repression and association with mRNA granules in response to acute glucose starvation. They show that Rvb1/Rvb2 associate with the target mRNAs in addition to their promoters, in a manner requiring only the gene promoter. Tethering Rvb1 or Rvb2 to a reporter mRNA is sufficient to repress its translation, stimulate its synthesis, and induce its sequestration in cytoplasmic granules; whereas Rvb2 knockdown eliminates the translational repression of several reporter mRNAs. These compelling findings support the important demonstration that Rvb1/Rvb2 are loaded on transcripts co-transcriptionally and accompany them into the cytoplasm where they repress translation in a manner associated with the accumulation of the repressed mRNAs in granules.

## Introduction

Gene expression encompasses many steps across discrete cellular boundaries, including transcription, mRNA processing and export, translation, and decay. Cells do not always live in stable and optimal

conditions, instead they are faced with various types of stresses, such as nutrient starvation, heat shock, toxins, pathogens, and osmotic stresses (*Majmundar et al., 2010*; *Richter et al., 2010*). In dynamic environmental conditions, cells must balance disparate responses in gene expression as they quickly transition between homeostatic states. This can present challenges such as when cells repress overall translation while needing to upregulate the protein expression of stress response genes (*de Nadal et al., 2011*). To date, it is generally thought that mRNA cytoplasmic activities are predominantly dictated by *cis*-acting sequence elements within the RNA; however, coupling steps in gene expression presents an attractive strategy to overcome the challenges by creating regulons of mRNAs that are similarly controlled at the transcriptional level and can be coordinately tuned at the post-transcriptional level as well.

In recent years. 'imprinting' by co-transcriptional loading has been implicated as an alternative mechanism to *cis*-acting RNA sequence elements in determining cytoplasmic mRNA fate (*Choder, 2011*; *Haimovich et al., 2013*). For instance, it was found that promoters determined mRNA decay rates through the co-transcriptional loading of RNA-binding proteins (RBPs) to the nascent RNA (*Bregman et al., 2011*; *Trcek et al., 2011*). Similarly, Vera et al. showed that the translation elongation factor eEF1A coupled the transcription and translation of *HSP70* mRNAs through co-transcriptional loading during heat shock in mammalian cells (*Vera et al., 2014*). Zander et al. showed that transcription factor Hsf1 might function in loading the nuclear mRNA export protein Mex67 on stress-related mRNAs during heat shock in yeast (*Zander et al., 2016*).

During stressful conditions, one proposed means of post-transcriptional control is the phase separation of select mRNA transcripts and post-transcriptional regulatory proteins into phase-dense, concentrated, and membrane-less cytoplasmic structures generally described as phase-separated granules (*Glauninger et al., 2022*; *Guzikowski et al., 2019*; *Zid and O'Shea, 2014*). Two well-known stress-induced phase-separated messenger ribonucleoprotein (mRNP) granules are processing bodies (P-bodies) and stress granules (*Protter and Parker, 2016*; *Youn et al., 2019*). During stress, the direct connection between the formation of these granules coincident with an overall translational reduction suggests that the localization of mRNAs to these cytoplasmic granules might sequester the mRNAs away from the translational machineries, thus repressing the translation of the mRNAs (*Attwood et al., 2020*; *Ivanov et al., 2019*; *Kedersha and Anderson, 2002*; *Sahoo et al., 2018*). Yet how mRNAs are partitioned to or excluded from stress-induced granules remains unclear.

Previously we found that during glucose starvation in yeast, promoter sequences play an important role in determining the cytoplasmic fate of mRNAs (*Zid and O'Shea, 2014*). mRNAs transcribed by active promoters in unstressed cells (class III, e.g., *PGK1*, *PAB1*) were directed to P-bodies and are poorly translated (*Guzikowski et al., 2022*). Meanwhile, stress-induced mRNAs showed two distinct responses: mRNAs of most heat shock genes (class I, e.g., *HSP30, HSP26*) are transcriptionally induced, actively translated, and remain diffuse in the cytoplasm; however, class II mRNAs are transcriptionally induced but become sequestered in both P-bodies and stress granules and are associated with inactive translation. Class II mRNAs are enriched for alternative glucose metabolic function such as glycogen metabolism (e.g., *GSY1*, *GLC3*, *GPH1*). Surprisingly, instead of the mRNA sequence itself, the promoter sequence that sits in the nucleus directs the translation and cytoplasmic localization of the corresponding induced mRNAs. Specifically, Hsf1-target sequences were shown to direct mRNAs to be excluded from mRNP granules and well translated. However, the mechanism by which the promoter can couple steps of gene expression during glucose starvation is unclear. As the promoter exclusively resides in the nucleus, we hypothesize factors exist that interact with promoters and are co-transcriptionally loaded onto mRNA prior to nuclear export.

In this study, we developed a novel proteomics-based screening method that enabled us to identify Rvb1/Rvb2 as interacting proteins with the promoters of the class II alternative glucose metabolism genes (e.g., *GLC3*) that are upregulated in transcription but downregulated in translation and have granular-localized mRNA transcripts. Rvb1/Rvb2 (known as RuvbL1/RuvbL2 in mammals) are two highly conserved AAA+ (ATPases Associated with various cellular Activities) proteins that are found in multiple nucleoprotein complexes. Structural studies have shown that in yeast they form an alternating heterohexameric ring or two stacked heterohexameric rings (*Jeganathan et al., 2015*). They were reported as the chaperones of multiprotein complexes involved in chromatin remodeling processes and other nuclear pathways including snoRNP assembly (*Eickhoff and Costa, 2017*; *Huen et al., 2010*; *Jeganathan et al., 2015*; *Jha and Dutta, 2009*; *Nano and Houry, 2013*; *Paci et al.,*

*2012*; *Seraphim et al., 2021*; *Tian et al., 2017*). These two proteins are generally thought to act on DNA but have been found localized to cytoplasmic granules under stress and to be core components of mammalian and yeast cytoplasmic stress granules (*Jain et al., 2016*; *Kakihara et al., 2014*; *Rizzolo et al., 2017*). Rvb1/Rvb2 have also been shown to regulate the dynamics and size of stress granules (*Narayanan et al., 2019*; *Zaarur et al., 2015*). The dual presence of Rvb1/Rvb2 at chromatin and stress granules hints to their potential in coupling activities in the nucleus and cytoplasm. Furthermore, a human homolog of Rvb2 was found to be an RNA-binding protein that promotes the degradation of translating HIV-1 *Gag* mRNA (*Mu et al., 2015*). Relatedly, in this study we found that Rvb1/Rvb2 have roles in coupling transcription, cytoplasmic mRNA localization, and translation of specific glucose starvation-induced genes in yeast, providing insight into how gene expression can be coordinated during fluctuating environmental conditions.

## Results

### Rvb1/Rvb2 co-purify with plasmids containing an alternative glucose metabolism gene promoter

To identify proteins involved in the ability of promoter sequences to direct the cytoplasmic fate of mRNAs during stress, we developed Co-Transcriptional ImmunoPrecipitation (CoTrIP), a novel biochemical screening technique to identify co-transcriptionally loaded protein factors (*Figure 1A*). Here, we modified a yeast plasmid containing LacO-binding sites that were previously used as an efficient purification system to isolate histones (*Unnikrishnan et al., 2010*; *Unnikrishnan et al., 2012*). To this plasmid, we added a uniform cyan fluorescent protein (CFP) open-reading frame (ORF) and different promoters of interest. We then used FLAG-tagged LacI, which binds to the LacO sequences, and UV-crosslinking to purify the plasmid along with the nascent mRNAs, and co-transcriptionally loaded proteins. Thereafter, mass spectrometry was performed to identify proteins enriched in a promoter-specific manner. Real-time quantitative PCR (RT-qPCR) validates that the CoTrIP method yields enrichment of the target nascent mRNAs, indicating that proteins enriched could be co-transcriptionally loaded (*Figure 1—figure supplement 1*). Here, we performed CoTrIP of three plasmids (two heat shock genes' promoters, *HSP30* and *HSP26*, and an alternative glucose metabolism gene's promoter, *GLC3*) in cells subject to 10 min of glucose deprivation. Those promoters had previously been shown to be sufficient to determine the cytoplasmic fate of the uniform ORF (*Zid and O'Shea, 2014*).

After comparing the protein enrichment on *GLC3* promoter and on *HSP30/HSP26* promoters (*Figure 1B and C*), we were able to detect differences in protein factors across the specific classes of promoters. The ATP-dependent DNA RuvB-like helicase Rvb1 was enriched tenfold more on *GLC3* promoter plasmids versus both *HSP30/HSP26* promoters (p-value=0.02). To further verify this enrichment, we compared our protein enrichment data against the CRAPome repository, a large database of contaminant proteins from various immunoprecipitation (IP) experiments, and we found that Rvb1 was significantly enriched on the *GLC3* promoter-containing plasmid (*Figure 1C*; *Mellacheruvu et al., 2013*). Proteins that were both enriched in 'promoter versus promoter comparison' as well as in comparison to the CRAPome are listed (*Figure 1C*).

Rvb1/Rvb2 are two highly conserved members of the AAA+ family that are involved in multiple nuclear pathways (*Jha and Dutta, 2009*). These two proteins are generally thought to act on DNA but have been found to be core components of mammalian and yeast cytoplasmic stress granules (*Jain et al., 2016*; *Kakihara et al., 2014*; *Rizzolo et al., 2017*). Microscopy revealed that Rvb1/Rvb2 are predominately present in the nucleus when cells are not stressed but a portion of them becomes localized to cytoplasmic granules that are distinct from P-bodies after 30 min glucose starvation conditions (*Figure 1—figure supplement 2*). Similar results were previously seen with 2-deoxyglucose-driven glucose starvation, where Rvb1 formed cytoplasmic foci independent of P-bodies and stress granules (*Rizzolo et al., 2017*). Rvb1/Rvb2's interactions with DNA in the nucleus and presence in the cytoplasm suggest the potential of Rvb1/Rvb2 to shuttle between the nucleus and cytoplasm.

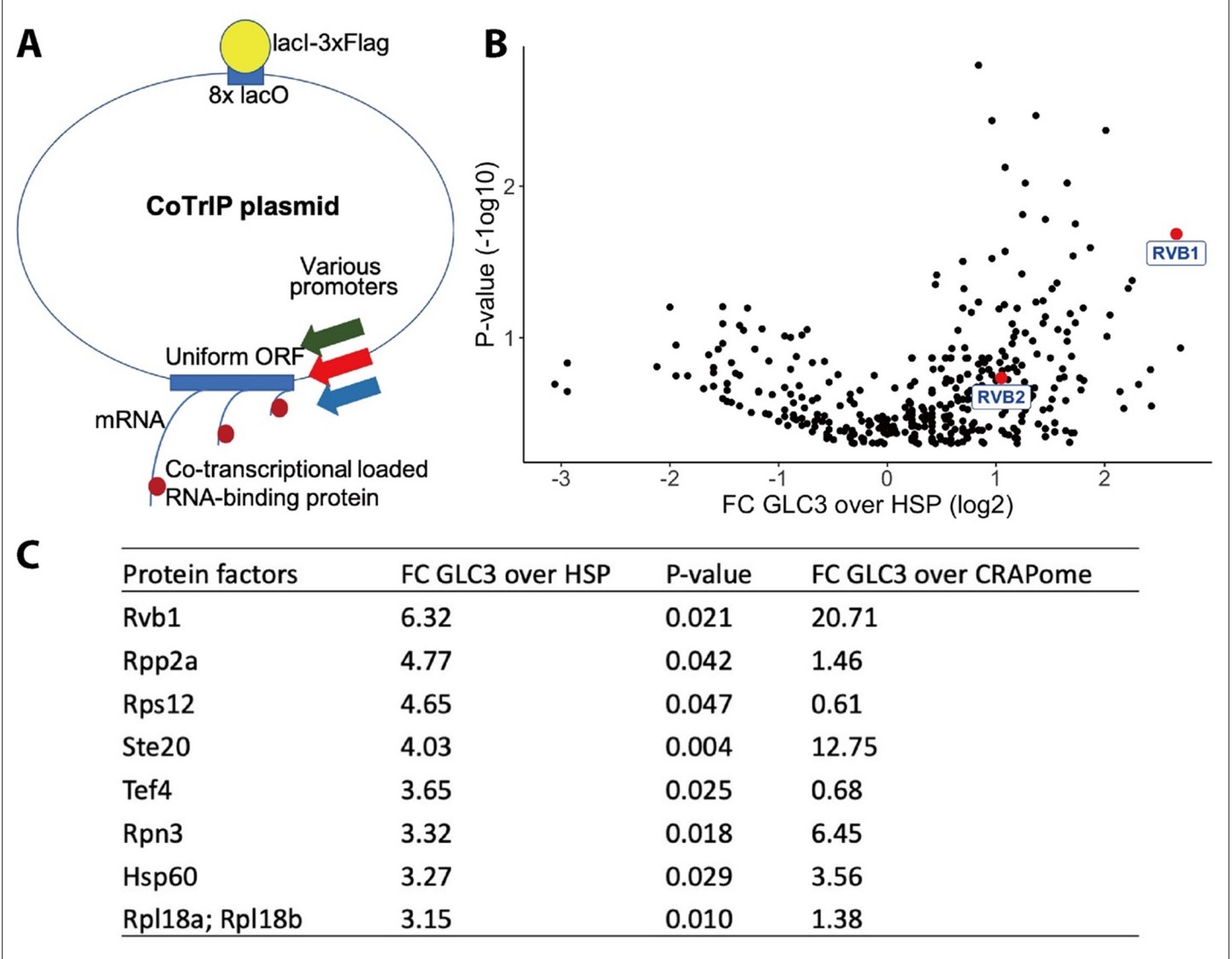

**Figure 1.** Rvb1/Rvb2 are identified as potential co-transcriptional-loaded protein factors on alternative glucose metabolism genes. (**A**) A schematic view of Co-Transcriptional ImmunoPrecipitation (CoTrIP). CoTrIP plasmid has an 8X lacO, a uniform open-reading frame (ORF), and various promoters of interest. CoTrIP plasmid was purified by immunoprecipitation of lacI-3XFlag protein. Enriched protein factors were identified by mass spectrometry. (**B**) Quantitative volcano plot of co-transcriptional-loaded protein candidates. X-axis: log2 scale of fold change of protein enrichment on two replicates of *GLC3* promoter-containing CoTrIP plasmid over on two replicates of *HSP30* promoter-containing and one replicate of *HSP26* promoter-containing plasmid. Y-axis: minus log10 scale of the p-values from two-sample *t*-test. Null hypothesis: enrichment on *GLC3* promoter equals the enrichment on *HSP* promoters. Rvb1 and Rvb2 are highlighted in red dots and labeled. (**C**) Table of protein factors enriched on *GLC3* promoters. FC of *GLC3* vs. *HSP*: fold change of protein enrichment on two replicates of *GLC3* promoter-containing CoTrIP plasmid over on two replicates of *HSP30* promoter-containing and one replicate of *HSP26* promoter-containing plasmid. GFPFC of *GLC3* vs. CRAPome: fold change of protein enrichment on two replicates of *GLC3* promoter-containing CoTrIP plasmid over the CRAPome repository. CRAPome: a contaminant repository for affinity purification–mass spectrometry data. CRAPome was used as a negative control. p-Values were from two-sample *t*-test of GLC3 vs. HSP. Null hypothesis: enrichment on *GLC3* promoter equals the enrichment on *HSP* promoters. Protein factors were ranked from highest to lowest by 'FC *GLC3* over *HSP*.'.

The online version of this article includes the following source data and figure supplement(s) for figure 1:

**Source data 1.** Data of fold change of enrichment on GLC3 over HSP promoters and P-values for enriched proteins.

**Figure supplement 1.** Reporter RNA was enriched upon Co-Transcriptional ImmunoPrecipitation (CoTrIP) plasmid immunoprecipitation.

**Figure supplement 1—source data 1.** Data points of normalized enrichment of reporter mRNA upon Co-Transcriptional ImmunoPrecipitation (CoTrIP).

**Figure supplement 2.** Rvb1/Rvb2 form cytoplasmic granules that are not co-localized with P-body during glucose starvation.

## Rvb1/Rvb2 are enriched at the promoters of endogenous alternative glucose metabolism genes

To validate the CoTrIP results as well as more globally explore the location of Rvb1 and Rvb2 on DNA during stress, Chromatin ImmunoPrecipitation sequencing (ChIP-seq) was used to investigate Rvb1/Rvb2's enrichment across the genome. Rvb1/Rvb2 were fused with a tandem affinity purification (TAP)-tag at the C-terminus and purified by rabbit IgG beads. The TAP-tagged strains grow at a normal rate (~90 min doubling time), which suggests TAP-tagging does not generally disrupt the endogenous protein function of these essential proteins. Here, we performed ChIP-seq on Rvb1, Rvb2, and the negative control Pgk1 in 10 min of glucose starvation (the Western validation of Rvb1 and Rvb2's IP is shown in *Figure 2—figure supplement 1A*). Rvb1/Rvb2 are enriched from the –500 bp to the transcription start site (TSS) along the genome at 10 min of glucose starvation, whereas ChIP-seq of the negative control Pgk1 is not enriched in the promoter region (*Figure 2A*). The overall enrichment of promoters is consistent with findings that Rvb's can function as chromatin remodelers (*Zhou et al., 2017*). We found that Rvb1/Rvb2 are highly enriched on *GSY1*, *GLC3,* and *HXK1* promoters but not *HSP30*, *HSP26,* or *HSP104* promoters, which is consistent with our CoTrIP results (*Figure 2C*). Rvb1/Rvb2 are significantly more enriched on the proximal promoters of the transcriptionally upregulated, poorly translated genes versus the transcriptionally upregulated and well-translated genes and the average genome (*Figure 2D*, *Figure 2—source data 1*). More generally we found that, for genes that show a greater than threefold increase in mRNA levels during glucose starvation, their promoters are significantly more enriched for Rvb2 binding. Previously we had found that Hsf1-binding sequences were sufficient to exclude mRNAs from mRNP granules during glucose starvation (*Zid and O'Shea, 2014*). Interestingly we found that glucose starvation-induced Hsf1-target promoters have no difference in Rvb1/Rvb2 binding than an average gene, and significantly lower Rvb1/Rvb2 enrichment than stress-induced non-Hsf1 targets (*Figure 2B*).

Enrichment peaks of Rvb1/Rvb2 were called using the macs algorithm (*Zhang et al., 2008*). Consistently, enrichment peaks of Rvb1/Rvb2 were identified on the promoter regions of the class II alternative glucose metabolism genes but not the class I heat shock genes (*Figure 2—source data 2*). Rvb1 and Rvb2 also show a highly overlapped enrichment pattern across the genome, but neither of them shows overlapped enrichment with the negative control Pgk1 (*Figure 2—figure supplement 1C*). Structural studies have shown that Rvb1/Rvb2 assemble as heterohexamers (*Gribun et al., 2008*). Their overlapped ChIP enrichment further supports that Rvb1 and Rvb2 function together along DNA.

## Rvb1/Rvb2 are co-transcriptionally loaded on the alternative glucose metabolism mRNAs

Although Rvb1/Rvb2 are predominantly considered to act on DNA, they are also found to interact with various mRNAs and regulate mRNA translation and stability (*Izumi et al., 2012*; *Mu et al., 2015*). We next sought to test whether Rvb1/Rvb2 established similar enrichment patterns on mRNAs. To test the interaction, we performed RNA ImmunoPrecipitation (RIP) on Rvb1, Rvb2, and the negative control wild-type (WT) strain followed by RT-qPCR in both log-phase and 15 min glucose-starved cells. Consistently, during 15 min of glucose starvation, Rvb1/Rvb2 are significantly more enriched on the mRNAs of the class II alternative glucose metabolism genes versus the class I heat shock genes (*Figure 3A*). Rvb2 is specifically highly enriched on *GSY1* mRNA, where it is around 20-fold more enriched than on *HSP30* mRNAs. However, in glucose-rich log-phase conditions, Rvb1/Rvb2 are generally less enriched on the mRNAs compared to starvation conditions. Additionally, in log phase, Rvb1/Rvb2 do not show differential enrichment between the alternative glucose metabolism genes and the heat shock genes (*Figure 3—figure supplement 1*).

Since Rvb1/Rvb2 are enriched on both promoters and mRNAs of class II alternative glucose metabolism genes, we hypothesized that Rvb1/Rvb2 are loaded from the interacting promoters to the nascent mRNAs via the transcription process. To test this, we eliminated the effects from the ORF sequences by designing a pair of reporter mRNAs with a uniform *CFP* ORF but driven by either the *GLC3* promoter or *HSP26* promoter (*Figure 3B*). Interestingly, although the mRNA transcribed virtually identical mRNA sequences (*Zid and O'Shea, 2014*), Rvb1/Rvb2 are significantly more enriched on the mRNA driven by the *GLC3* promoter compared to the one driven by the *HSP26* promoter during 15 min of glucose starvation (*Figure 3C*). This suggests that only the promoter itself can determine

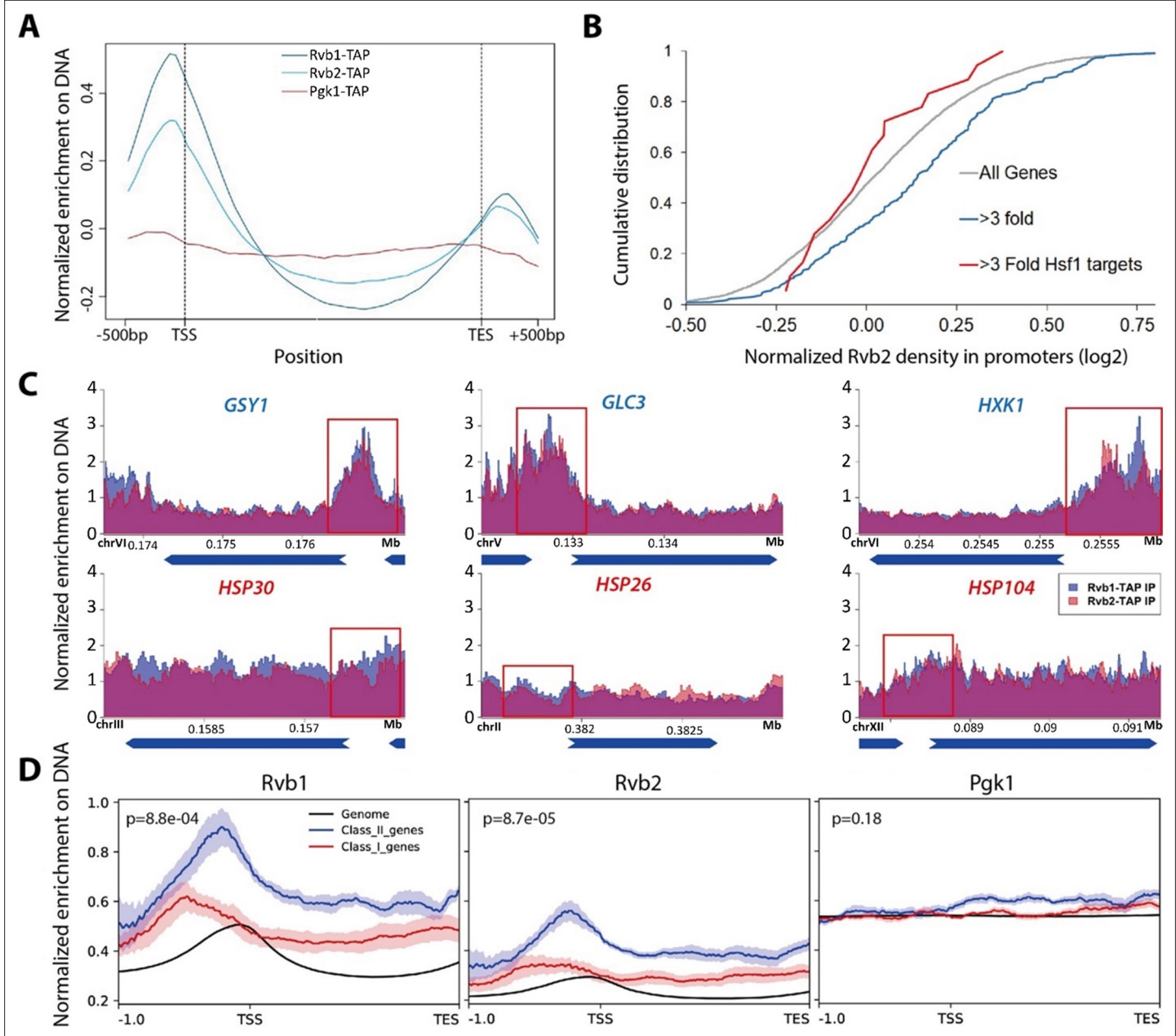

**Figure 2.** Rvb1/Rvb2 are enriched at the promoters of endogenous alternative glucose metabolism genes. (**A**) Rvb1/Rvb2 are enriched on promoters and nascent gene bodies. Chromatin ImmunoPrecipitation sequencing (ChIP-seq) of cells in 10 min glucose starvation. X-axis: normalized scale of all genes containing −500 bp to transcription start site (TSS), TSS to transcription end site (TES) and TES to +500 bp. Y-axis: normalized density of target protein on the loci. Normalized density: RPKM of ChIP over RPKM of input. Input: 1% of the cell lysate. Rvb1/Rvb2/Pgk1 are C-terminally fused with tandem affinity purification (TAP) tag and immunoprecipitated by IgG-conjugating breads. Pgk1: a negative control that considered as noninteractor on the genome. (**B**) Cumulative distribution of Rvb2's enrichment on genes. X-axis: log2 scale of Rvb2 ChIP read counts over Pgk1 ChIP read counts from −500 bp to TSS. Y-axis: cumulative distribution. >3-fold: genes that have more than threefold transcriptional induction during 10 min glucose starvation. >3-fold Hsf1 targets: genes that have more than threefold transcriptional induction and are Hsf1-regulated. List of genes is given in the supplementary file. (**C**) Representative gene tracks showing Rvb1/Rvb2's enrichment. X-axis: gene track with annotation (in Mb). Arrow's orientation shows gene's orientation. Y-axis: normalized density of Rvb1/Rvb2 over Pgk1. Normalized density: RPKM of ChIP over RPKM of input. Class I genes are labeled in red and class II genes are labeled in blue. Promoters are highlighted by red rectangles. (**D**) Enrichment profile of Rvb1/Rvb2 on class I, II genes and genome. X-axis: normalized scale of genome containing −500 bp to TSS, TSS to TES. Y-axis: RPKM of ChIP over RPKM of input. p-Values are from two-sample *t*-test. Null hypothesis: normalized density from −500 bp to TSS on class II promoters equals on class I promoters.

The online version of this article includes the following source data and figure supplement(s) for figure 2:

**Source data 1.** List of class I upregulated and high-ribo genes and class II upregulated and low-ribo genes.

*Figure 2 continued on next page*

*Figure 2 continued*

**Source data 2.** List of Rvb1/Rvb2 peak calls on the genome.

**Figure supplement 1.** Western blot validation of Chromatin ImmunoPrecipitation sequencing (ChIP-seq), and Rvb1 and Rvb2's enrichment regions are highly overlapped.

**Figure supplement 1—source data 1.** Rvb1 source data.

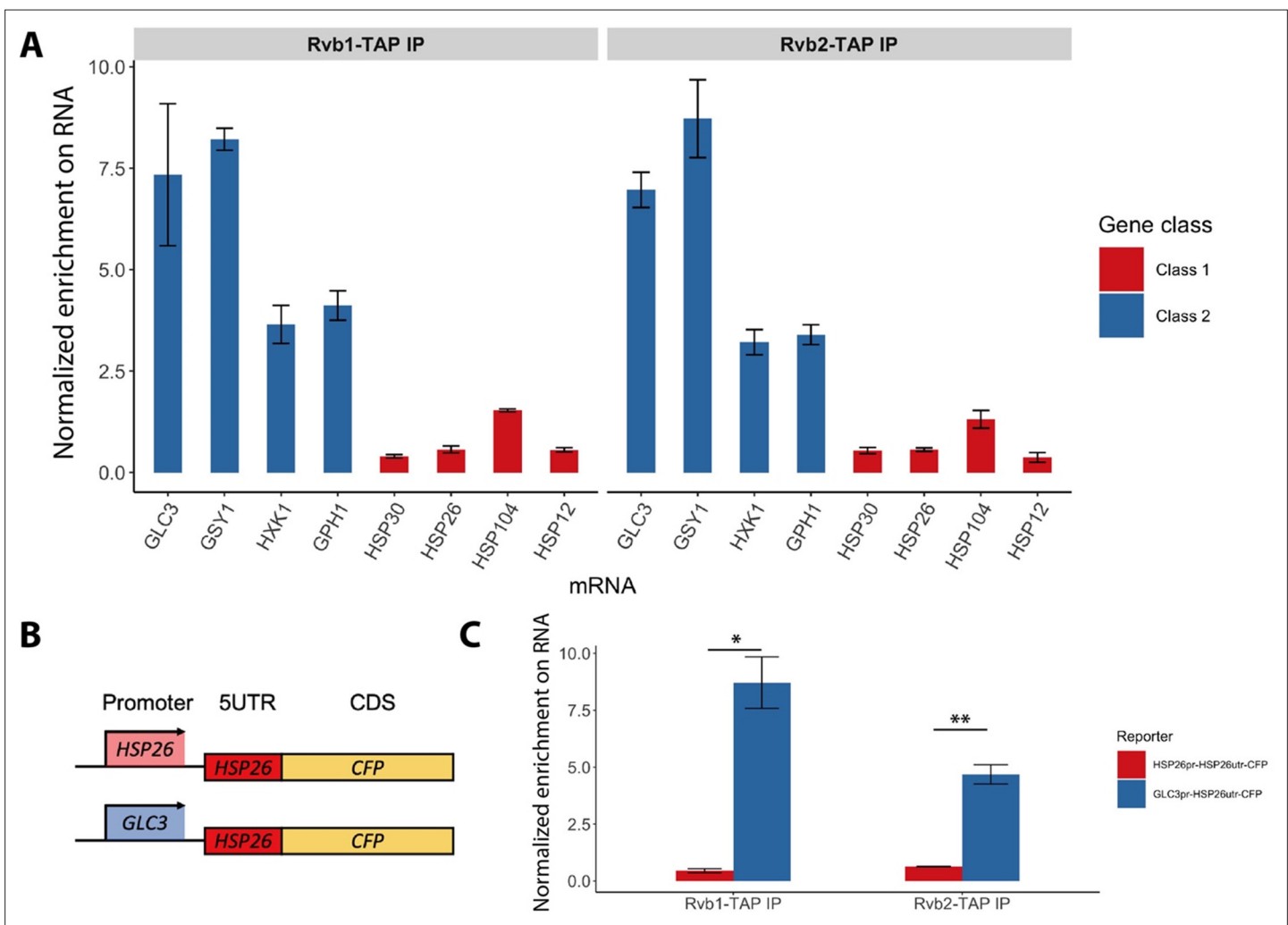

**Figure 3.** Rvb1/Rvb2 are co-transcriptionally loaded on the alternative glucose metabolism mRNAs. (**A**) Rvb1/Rvb2's enrichment on endogenous mRNAs in 15 min glucose starvation. RNA immunoprecipitation qPCR of cells in 15 min glucose starvation. Error bars are from two biological replicates. X-axis: four class I mRNAs labeled in red and four class II mRNAs in blue. Y-axis: Ct values were firstly normalized by internal control *ACT1*, then normalized by input control, finally normalized by the wild-type immunoprecipitation control group. Input: 1% of the cell lysate (n = 3). (**B**) A schematic view of the reporter mRNA only swapping the promoter. 5UTR: 5′ untranslated region; CDS: coding sequence; CFP: cyan fluorescent protein. (**C**) Rvb1/ Rvb2's enrichment on the reporter CFP mRNAs in 15 min glucose starvation. RNA immunoprecipitation qPCR of cells in 15 min glucose starvation. X-axis: *HSP26* promoter-driven reporter mRNA labeled in red and *GLC3* promoter-driven mRNA in blue. Y-axis: Ct values were firstly normalized by internal control *ACT1*, then normalized by input control. Input: 1% of the cell lysate. Standard deviations are from two biological replicates. Statistical significance was assessed by two-sample *t*-test (*$p < 0.05$, **$p < 0.01$). Null hypothesis: the enrichment on the two reporter mRNAs is equal.

The online version of this article includes the following source data and figure supplement(s) for figure 3:

**Source data 1.** Data of endogenous mRNA enrichment from Rvb1/Rvb2 pulldown in glucose starvation.

**Figure supplement 1.** Rvb1/Rvb2 did not show differential enrichment between class I and II mRNAs in glucose-rich log-phase cells.

**Figure supplement 1—source data 1.** Data of mRNA enrichment from Rvb1/Rvb2 pulldown on endogenous class I and II mRNAs in glucose-rich log-phase cells.

the transcribed mRNA's interaction with Rvb1/Rvb2, further indicating that Rvb1/Rvb2 are likely to be co-transcriptionally loaded from the promoters to nascent mRNAs.

## Engineered Rvb1/Rvb2 tethering to mRNAs directs the cytoplasmic localization and repressed translation

As Rvb1/Rvb2 were found to be located at both promoters in the nucleus and associated with mRNAs in the cytoplasm, we asked whether Rvb1/Rvb2 have an impact on the cytoplasmic fates of bound mRNAs. To test this, we engineered interactions between Rvb1 or Rvb2 and the mRNAs transcribed from various promoters of class I heat shock genes (e.g., *HSP30*, *HSP26*). We took advantage of the specific interaction between a phage-origin PP7 loop RNA sequence and the PP7 coat protein (*Lim and Peabody, 2002*). Here, in our engineered strains, a reporter construct consists of a promoter of interest, a nanoluciferase (nLuc) reporter ORF for measuring protein synthesis, a PP7 loop to drive the engineered interaction, and an MS2 loop for the mRNA subcellular visualization. Along with the reporter, Rvb1 or Rvb2 are fused with PP7-coat protein to establish binding on the reporter mRNA (*Figure 4A*). As previously shown, Rvb1/Rvb2 do not display strong binding on the promoters and mRNAs of class I heat shock genes (*Figures 2C and 3A*) (e.g., *HSP30*). Therefore, we specifically engineered the interaction between Rvb1 or Rvb2 and two types of mRNAs driven by the class I heat shock promoters (*HSP30/HSP26*). Strikingly, binding of both Rvb1 and Rvb2 alters the cytoplasmic fates of these class I heat shock mRNAs to be similar to the class II alternative glucose metabolism mRNAs (*Figure 4*, *Figure 4—figure supplements 1 and 2*). Taking *HSP30* promoter-driven reporter mRNA as an example, during glucose starvation the binding of Rvb1 or Rvb2 reduces protein synthesis by ~40% (*Figure 4B*). It is important to consider that final protein abundance is determined by both mRNA levels and translation. Interestingly, we observed an increase in mRNA abundance when Rvb2 is tethered to the reporter mRNA (*Figure 4B*). When the translational efficiency was normalized by the mRNA abundance, we were surprised to observe Rvb2 tethering reduces translational efficiency by greater than twofold during glucose starvation (*Figure 4B*). This change in gene expression was specific to Rvb2 as tethering a GFP PP7-coat protein control had no significant impact on protein levels, mRNA levels, or translational efficiency. Additionally, Rvb1/Rvb2 binding does not significantly repress the translational efficiency of mRNA in glucose-rich unstressed cells, indicating that Rvb1/Rvb2 has a more significant effect on mRNAs when mRNP granules have visibly formed (*Figure 4—figure supplements 1 and 2*).

Since the translation of the mRNAs bound by Rvb1/Rvb2 was reduced, we further visualized the subcellular localization of those mRNAs. Consistent with reduced translation, Rvb1/Rvb2 tethering significantly increases the granular localization of the heat shock mRNA reporters (*Figure 4C*, *Figure 4—figure supplements 1 and 2*). Taking *HSP30* promoter-driven reporter mRNA as an example, only 4% of the cells form *HSP30* promoter-driven mRNA-containing granules when the mRNA is not bound by Rvb1 or Rvb2, yet Rvb1 tethering increases the mRNA's granular localization to 27% of the cells and Rvb2 tethering increases the mRNA's granular localization to 39% of the cells (*Figure 4C and D*). Furthermore, the binding of Rvb1 and Rvb2 to mRNA increases the formation of granules that are non-colocalized with a P-body marker (*Figure 4—figure supplements 1 and 2*). This indicates that Rvb2 is sufficient to drive the interacting mRNA to P-body independent starvation-induced granules. To further eliminate any potential artifacts caused by the C-terminal modification on Rvb1/Rvb2, the negative controls were tested where Rvb1 or Rvb2 are fused with PP7 coat protein, but the mRNA does not have the PP7 loop. The negative control strains did not show a decrease in translatability of the reporter mRNAs in glucose starvation. It indicates translation only decreases and the mRNA granular localization level only increases when the full Rvb-mRNA interaction was established (*Figure 4—figure supplements 1 and 2*). These results support the ability of Rvb1/Rvb2 to suppress the translation of the binding mRNAs, potentially through sequestering the mRNAs into cytoplasmic granules.

The coupling of induced transcription and repressed translation of the class II alternative glucose metabolism genes may be an important adaptation for cells to survive from stress conditions. Results showed that after replenishing the glucose to the starved cells, the translation of those genes is quickly induced, with an approximately eightfold increase in ribosome occupancy 5 min after glucose readdition for class II mRNAs (*Figure 4—figure supplement 3A*). We also find that class II mRNAs quickly increase their protein production upon glucose readdition (*Figure 4—figure supplement*

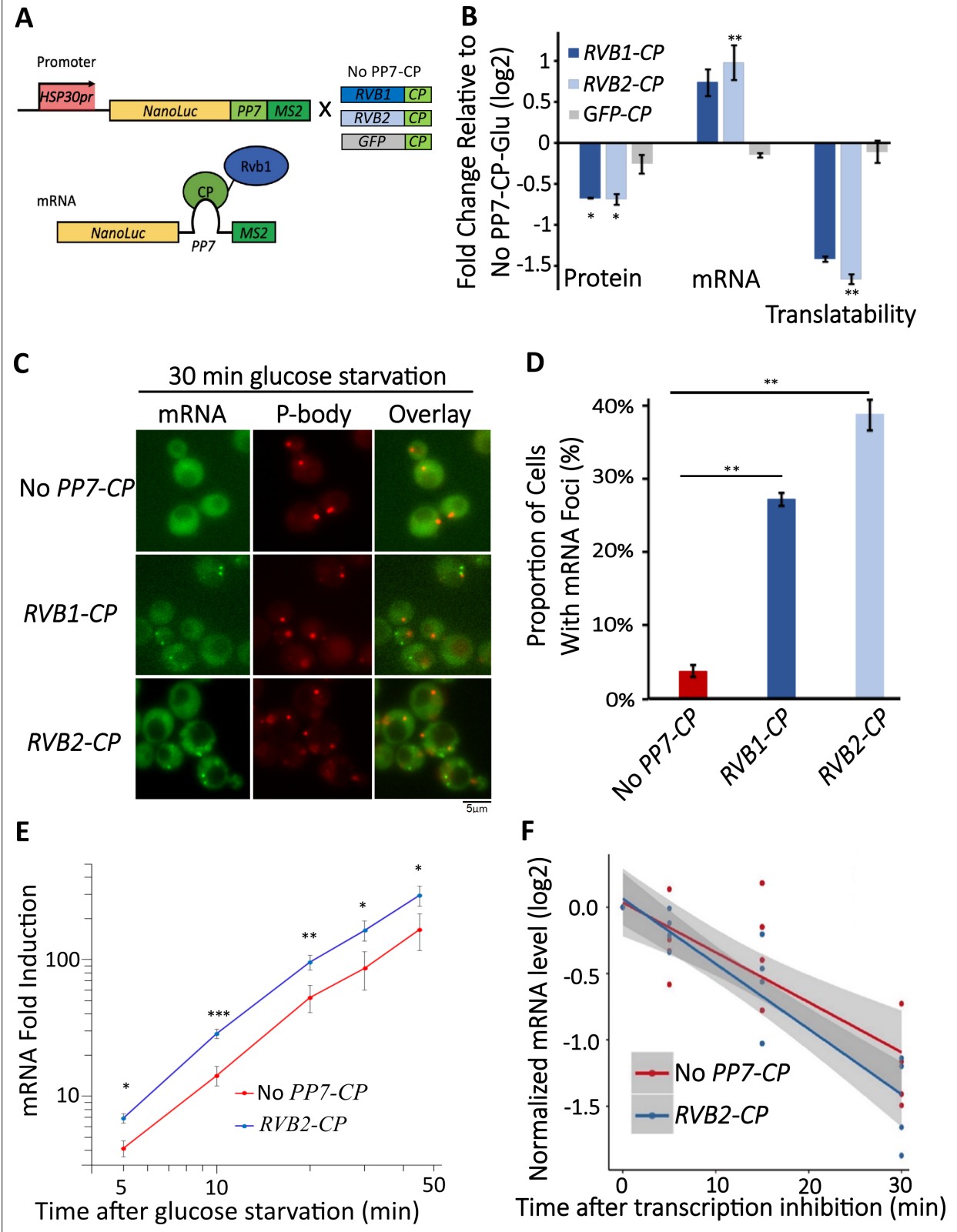

**Figure 4.** Engineered Rvb1/Rvb2 tethering to mRNAs directs cytoplasmic granular localization and repressed translation. (**A**) A schematic view of Rvb-tethering methodology. The reporter mRNA contains an HSP30 promoter, nLuc CDS, PP7 loop sequence, and 12XMS2 sequence. Rvb1, Rvb2, or GFP are C-terminally fused with PP7-coat protein (CP). Upper panel shows cloning strategy and lower panel shows mRNA's situation upon engineering. (**B**) Protein, mRNA, and translatability levels of *HSP30* promoter-driven reporter mRNA in glucose starvation. Y-axis: nLuc synthesized within 5 min time

*Figure 4 continued on next page*

*Figure 4 continued*

frame after 25 min glucose starvation. NLuc reading was subtracted by the nLuc reading of cycloheximide added 5 min earlier. mRNA levels of *HSP30* promoter-driven reporter mRNA in 15 min glucose starvation relative to No PP7-CP. Initial samples were normalized by the internal control *ACT1*. Translatability was calculated by subtracting the log2 protein values from mRNA relative to No PP7-CP. No PP7-CP (n = 5), RVB1-CP (n = 2), RVB2-CP (n = 5), and GFP-CP (n = 3). Error bars are SEM from these biological replicates. Statistical significance was assessed by a one-sample *t*-test to test whether the mean differs from 0 (no change from No PP7-CP) (*p<0.05, **p<0.01). (**C**) Live imaging showing the subcellular localization of the *HSP30* promoter-driven reporter mRNA in 30 min glucose starvation. Reporter mRNA is labeled by the MS2 imaging system. P-body is labeled by marker protein Dcp2. PP7 ctrl: negative control, cells only have the reporter mRNA with PP7 loop. PP7+Rvb1-PCP: Rvb1 is tethered to mRNA. PP7+Rvb2-CP: Rvb2 is tethered to mRNA. (**D**) quantification of the subcellular localization of the reporter mRNA.Y-axis: percentage of cells that have the reporter mRNA-containing granule foci (n = 200). Error bars are from two biological replicates. Statistical significance was achieved by two-sample *t*-test. Null hypothesis: the proportion of cells with mRNA foci mRNA of experimental and control groups is equivalent. (**E**) mRNA fold induction of Rvb2-tethered mRNAs and nontethered mRNAs over time. Reporter mRNA is *HSP30* promoter driven. X-axis: time (minute) after glucose is removed. Y-axis: mRNA fold induction compared to pre-stress condition (log scale). n = 4, error bars are the SEM of these four replicates. (**F**) mRNA decay curve of *HSP30* promoter-driven reporter mRNAs. X-axis: after cells were starved for 15 min, time (minute) after stopping the transcription using 1,10-phenanthroline. Y-axis: log2 scale of normalized mRNA levels. Ct values of reporter mRNAs were normalized by the internal control *ACT1*. Statistical significance was achieved by linear regression modeling. Null hypothesis: the mRNA levels of experimental and control groups are equivalent (*p<0.05, **p<0.01, **p<0.001).

The online version of this article includes the following source data and figure supplement(s) for figure 4:

**Source data 1.** Data of protein synthesis of Rvb1/Rvb2 tethering to HSP26 promoter-driven reporter mRNA and control mRNAs (non-tethered or GFP-tethered reporter mRNAs).

**Figure supplement 1.** Engineered Rvb1/Rvb2 tethering to *HSP30* promoter-driven reporter mRNA directs cytoplasmic granular localization and repressed translation.

**Figure supplement 1—source data 1.** Data of protein synthesis of Rvb1/Rvb2 tethering to *HSP30* promoter-driven reporter mRNA and control mRNAs.

**Figure supplement 2.** Engineered Rvb1/Rvb2 tethering to *HSP26* promoter-driven reporter mRNA directs cytoplasmic granular localization and repressed translation.

**Figure supplement 2—source data 1.** Data of protein synthesis of Rvb1/Rvb2 tethering to *HSP26* promoter-driven reporter mRNA and control mRNAs.

**Figure supplement 3.** Ribosome occupancy and protein synthesis of endogenous glucose metabolism mRNAs were quickly induced after glucose replenishment.

**Figure supplement 3—source data 1.** Data of ribosome occupancy of mRNAs in glucose starvation and in glucose replenishment.

**Figure supplement 4.** Engineered Rvb1/Rvb2 binding to mRNAs increases the transcription of corresponding genes.

**Figure supplement 4—source data 1.** Data of halflife measurements of Rvb2 tethering to HSP30 promoter-driven reporter mRNA and control mRNA.

*3B*). This indicates the potential biological role of the starvation-induced granules as a repository for these translationally repressed class II mRNAs during stress that does not preclude these mRNAs from potentially being quickly released and translated once the stress is removed.

## Engineered Rvb2 binding to mRNAs increases the transcription of corresponding genes

Interestingly, Rvb2 not only suppress the translation of bound mRNAs, but also increases the abundance of the interacting mRNAs by approximately almost twofold (*Figure 4C*). There are two possibilities for this increased mRNA abundance by Rvb2 tethering: increased transcription and/or slower mRNA decay. To address this, we performed time-course measurements on the *HSP30* promoter-driven reporter mRNA abundance in 0, 5, 10, 15, 30, and 45 min of glucose starvation. Here, we compared the mRNA abundance when mRNA is bound by Rvb2 and when mRNA is not bound by Rvb2 as a control. A mathematical modeling approach was performed to predict the mRNA induction abundance change caused by varied transcriptional efficiency or varied decay rate (*Figure 4—figure supplement 4*; *Elkon et al., 2010*). In the model, we assumed that mRNA level is mainly dependent on the transcriptional and decay rates, and these parameters stay constant over the course of induced expression during glucose starvation. From the mathematical modeling, mRNA fold induction differs between varied transcriptional induction versus mRNA decay changes. If transcriptional rates vary, differences in mRNA levels are the same (equal distance shift) at each time point on a log-log scale (*Figure 4—figure supplement 4*). While if mRNA decay varies, although little difference is seen in mRNA abundance in early glucose starvation, our simulation predicts increasing differences in mRNA abundance at later time points of glucose starvation (*Figure 4—figure supplement 4B*). By comparing experimental measurements of mRNA induction of the Rvb2-tethered condition to the

unbound control condition, the mRNA induction differences are similar at each time point. When Rvb2 binds to the mRNA, the abundance of the mRNA is constantly greater than the unbound mRNA at all time points during glucose starvation, indicating that the greater mRNA abundance is mainly due to greater transcription differences driven by the Rvb2 binding (*Figure 4E*).

To further experimentally validate that the greater mRNA abundance caused by Rvb2 binding is due to increased transcription and not slower decay, we stopped cellular transcription after 15 min of glucose starvation by treatment with the transcription inhibitor drug 1,10-phenanthroline. Then we performed time-course measurements on mRNA abundance and compared the decay of mRNAs with and without Rvb2 binding. Consistently, mRNAs bound by Rvb2 decay at a slightly but not significantly faster rate, not a slower rate (*Figure 4F*). Whether or not bound by Rvb2, the reporter mRNA has around a 25 min half-life (*Figure 4—figure supplement 4C*). These results further point to Rvb2 mRNA tethering driving transcriptional upregulation. Since Rvb2 is targeted to the mRNA, it is likely that local recruitment of Rvb2 to the nascently transcribed mRNA increases the local concentration of Rvb2 protein to the vicinity of the regulatory region of the corresponding gene, further showing the connections between the transcriptional and translational processes.

## RVB2 knockdown drives decreased mRNA induction but enhanced protein production of Rvb1/Rvb2 target genes

To further test the role of RVB1/RVB2 in regulating gene expression during glucose starvation, we sought to reduce RVB function in cells. RVB1/RVB2 deletions are inviable (*Jónsson et al., 2001*) so we aimed to identify gRNA targets that would temporally reduce RVB expression through inducible gRNAs and dCas9-MXi (*Smith et al., 2016*). We found an RVB2 gRNA that gave an ~20-fold reduction in RVB2 expression 8 hr after treatment with anhydrotetracycline (ATc) (*Figure 5A and B*). To investigate the necessity of Rvb1/Rvb2 in the translational repression of class II genes during glucose starvation, we C-terminal tagged two class II genes (*GSY1* and *HXK1*) and one class I gene (*HSP30*) and then quantified their protein induction after 30 min glucose starvation. While both class II genes had robust mRNA induction upon glucose starvation in the control samples (*Figure 5—figure supplement 1*), this was associated with no significant upregulation of protein production (*Figure 5C*). Upon *RVB2* knockdown, we find a significant increase in the stress induction of the class II proteins Gsy1 and Hxk1, while we find no significant difference in the protein induction of the class I protein Hsp30 (*Figure 5C*). While the higher protein induction could be because of even further increases in mRNA levels, we instead find that *RVB2* knockdown causes a greater than twofold decrease in *GSY1* and *HXK1* mRNA induction (*Figure 5D*). This is consistent with previous findings (*Jónsson et al., 2001*) as well as our tethering data that Rvb1/Rvb2 have a role in transcriptional induction. Together this data further supports the role for Rvb1/Rvb2 repressing the translatability of target mRNAs during glucose starvation.

## Discussion

In fluctuating environments, cells must quickly adjust the expression of different genes dependent upon cellular needs. Here, our results demonstrate a novel function of the AAA+ATPases Rvb1/Rvb2 in the cytoplasm, and a novel mechanism of Rvb1/Rvb2 in coupling the transcription, mRNA cytoplasmic localization, and translation of specific genes (*Figure 6*). We identified Rvb1/Rvb2 as enriched protein factors on the promoters of the class II alternative glucose metabolism genes that are upregulated in transcription but downregulated in translation during glucose starvation. Results showed that Rvb1/Rvb2 have a strong preferred interaction with both promoters and mRNAs of these genes, suggesting that Rvb1/Rvb2 are loaded from enriched promoters to the nascent mRNAs. More interestingly, when we tethered Rvb1/Rvb2 to the mRNAs, the binding of Rvb1/Rvb2 had a strong impact on reducing mRNA translation and increasing the mRNA granular localization. We are uncertain whether Rvb1/Rvb2 tethering represses translation, which directs mRNAs to mRNP granules; or Rvb1/Rvb2 binding directly targets the mRNA to the granule, which represses translation; or some combination of both – as these are very hard to disentangle. Either way, these data, along with our RVB2 depletion data, suggest the potential co-transcriptional loading of Rvb1/Rvb2 directs post-transcriptional mRNA fate in the cytoplasm. Additionally, Rvb1/Rvb2's interaction with the mRNA can also induce transcription of

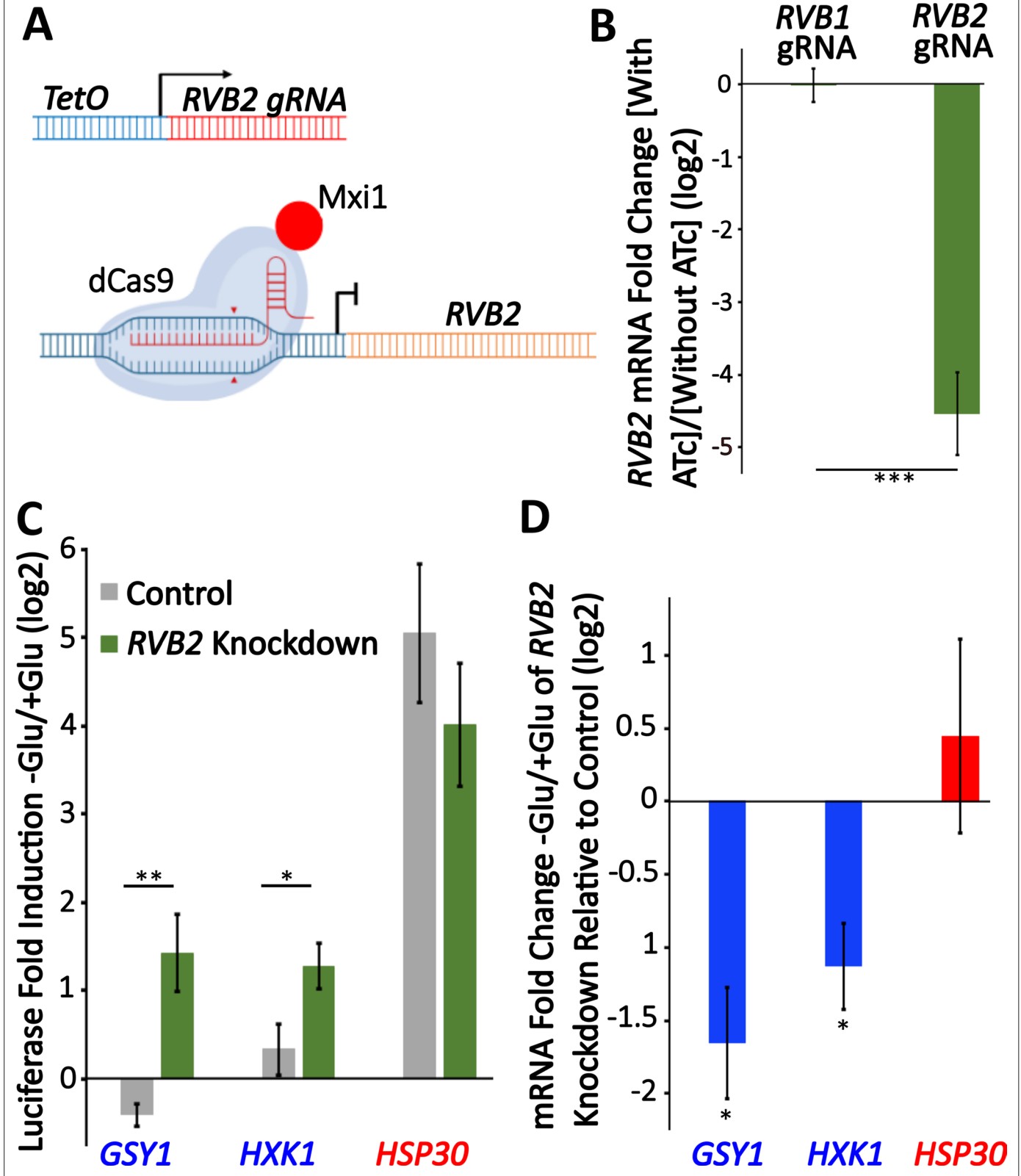

**Figure 5.** Knockdown of RVB2 drives enhances protein production of Rvb target genes during glucose starvation. (**A**) A schematic view of CRISPRi repression of *RVB2* transcription. The *RVB2 gRNA* was placed under the control of a TetOn promoter. Upon anhydrotetracycline (ATc) treatment, this induces *RVB2 gRNA* expression, targeting dCas9-MXi to the upstream region of *RVB2* and repressing transcription. (**B–D**) RVB2 knockdown was accomplished by inoculating log-phase cultures with 250 ng/L ATc for 8 hr. Control cells have no ATc. (**B**) mRNA levels of *RVB2* and *ACT1* were

*Figure 5 continued on next page*

*Figure 5 continued*

determined in log-phase cultures expressing either an *RVB1* or *RVB2* gRNA. *RVB2* mRNA levels were normalized to *ACT1*. Statistical significance was achieved by two-sample *t*-test. *RVB1 gRNA* (n = 2); *RVB2 gRNA* (n = 5). (**C**) Endogenous genes were tagged with nLuc and luciferase was quantified during log-phase growth and 30 min after glucose starvation in ±ATc cultures. Statistical significance was achieved by two-sample *t*-test (GSY1, HXK1 n = 4, HSP30 n = 3). (**D**) mRNA levels of the genes of interest were tested in log phase and 30 min of glucose starvation ±ATc. The log2 fold change -Glu/+Glu in the RVB2 knockdown was subtracted from the control mRNA fold change. Statistical significance was assessed by a one-sample *t*-test to test whether the mean fold change differs from 0 (no change from -ATc control) (GSY1, HXK1 n = 4, HSP30 n = 3) (*p<0.05, **p<0.01, **p<0.001).

The online version of this article includes the following source data and figure supplement(s) for figure 5:

**Source data 1.** Data of induction of RVB2 mRNA level with RVB1 and RVB2 gRNA, respectively.

**Figure supplement 1.** mRNA induction upon glucose starvation in RVB2 knockdown strains.

the corresponding genes, further indicating that Rvb1/Rvb2 couple the transcription and translation of the interacting genes.

It is not clear how tethering Rvb1/Rvb2 to an mRNA reporter increases transcription of the corresponding DNA locus. Rvb1/Rvb2 were initially found to be associated with many chromatin remodeling and transcription-related complexes. This has further been expanded on, and several studies have demonstrated a chaperone-like activity in the formation of various complexes including the assembly of chromatin remodeling complexes and RNA polymerase II (*Nano and Houry, 2013*; *Paci et al., 2012*; *Seraphim et al., 2021*). It may be that recruiting Rvb1/Rvb2 to the nascent RNA increases the local concentration, driving further enhancement of transcription-related processes. It is also intriguing to think about Rvb1/Rvb2's reported role in escorting client proteins to large macromolecular complexes. Like their escorting protein function, it is plausible to hypothesize that they might have additional functions in escorting mRNAs to large macromolecule mRNP granule complexes in stressful conditions.

The coupling of transcription and translation of specific genes may be an important adaptation for cells to survive during stress conditions. It has been postulated that to save energy during stress mRNAs are temporarily stored in the cytoplasmic granules associated with inactive translation instead of mRNA decay (*Guzikowski et al., 2019*; *Horvathova et al., 2017*; *Pitchiaya et al., 2019*; *Protter and Parker, 2016*; *Schütz et al., 2017*). Our results show that, after replenishing glucose to the starved cells, the translation of those genes is quickly induced (*Figure 4—figure supplement 3*).

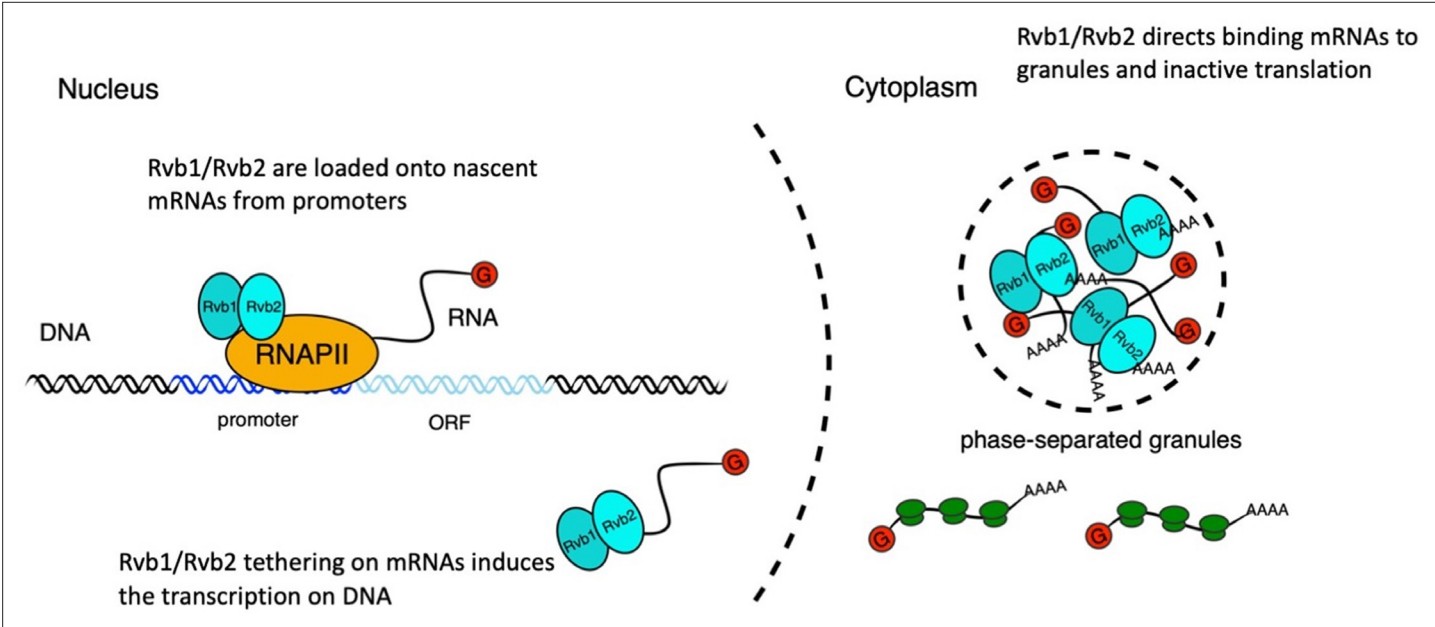

**Figure 6.** A working illustration of Rvb1/Rvb2's mechanism in coupling the transcription and translation of interacting genes. First, Rvb1/Rvb2 are recruited by specific promoters and loaded onto the nascent mRNAs during glucose starvation. Then Rvb1/Rvb2 escort the interacting mRNAs to the cytoplasm and cause repressed translation and localization to cytoplasmic granules. Also, forced Rvb binding on an mRNA drives an increase in the transcription of the corresponding genes, further showing the coupling of transcription and translation.

The stress-induced phase-separated granules may serve as temporary repositories for the inactive translating mRNAs of many genes that are regulated by Rvb1/Rvb2 in glucose starvation and involved in alternative glucose metabolism pathways. From the perspective of cell needs, many alternative glucose metabolism genes are involved in glycogen synthesis, which may be superfluous for survival during times of complete glucose starvation, but the cell may want to produce them as quickly as possible upon glucose replenishment to drive quick protein synthesis. The special coupling of increased transcription but repressed translation mediated by Rvb1/Rvb2 may serve as an emergency but prospective mechanism for cells to precisely repress the translation of these alternative glucose metabolism mRNAs during stress but be able to quickly translate these pre-stored mRNAs once the cells are no longer starved (*Jiang et al., 2020*). Also, the genes regulated by Rvb1/Rvb2 may be dependent on the type of stresses. We observed this mechanism of gene expression control on alternative glucose metabolism genes during glucose starvation stress. It will be interesting to test whether there is similar regulation on different sets of genes that are related to other types of stresses, such as heat shock and osmotic stress responses.

To further understand the function of Rvb1/Rvb2 mechanistically, it is crucial to understand how Rvb1/Rvb2 are recruited to these specific promoters. We prefer the hypothesis that the recruitment of Rvb1/Rvb2 is mediated by other DNA-binding proteins as we were unable to identify specific binding motifs of Rvb1/Rvb2 from our ChIP-seq data. As we found that Rvb1/Rvb2 are generally enriched on the promoters of transcriptionally upregulated mRNAs, we favor a model in which the default is for Rvb1/Rvb2 to be recruited to active transcription sites. This fits with previous data that Rvb1/Rvb2 are required to maintain expression of many inducible promoters including galactose-inducible transcripts (*Jónsson et al., 2001*). While Rvb1/Rvb2 are generally recruited to the promoters of induced mRNAs during glucose starvation, we find that Hsf1-regulated promoters circumvent this recruitment through an unknown mechanism as the transcriptionally upregulated Hsf1 targets show reduced recruitment relative to non-Hsf1 targets (*Figure 2B*). Intriguingly, Hsf1-regulated genomic regions have been found to coalesce during stressful conditions (*Chowdhary et al., 2017*; *Pincus et al., 2018*). It will be interesting to explore whether Rvb1/Rvb2 may be excluded from these coalesced regions in future studies.

This study provides new insights into how gene expression is controlled under stressful conditions, including how mRNAs can be targeted to stress-induced mRNP granules. It also identifies Rvb1/Rvb2 as key proteins connecting discrete steps of gene expression across cellular compartments. In mammalian cells, overexpression of the *RVB1/2* homologs, *RUVBL1/2*, is correlated with tumor growth and poor prognosis in several cancer types, yet precise mechanisms for how these proteins impact cancer progression are unclear (*Grigoletto et al., 2011*; *Lauscher et al., 2012*; *Lin et al., 2020*). It is important to further study the role these proteins have on connecting gene expression in different conditions to better understand how they may be impacting cancer progression in mammalian cells.

## Materials and methods
### Yeast strains and plasmids
The yeast strains and plasmids used in this study are listed in Key resources table, and the oligonucleotides used for the plasmid construction, yeast cloning, and RT-qPCR are described in Key resources table. The strains were created through genomic integration of a linear PCR product, or a plasmid linearized through restriction digest or the transformation of an episomal vector. The background strain used was W303 (EY0690), one laboratory strain that is closely related to S288C. In yeast cloning for the C-terminal fusion on the endogenous proteins (e.g., Rvb1-mNeongreen, Rvb1-PP7CP, Rvb1-TAP), we used plasmids of Pringle pFA6a and pKT system (*Lee et al., 2013*; *Longtine et al., 1998*; *Zid and O'Shea, 2014*), gifts from the E. K. O'Shea laboratory and the K. Thorn laboratory. We modified the pFA6a and pKT plasmids by inserting in the peptides of interest into the plasmids. The primers used to amplify the fragments from these plasmids contain two parts from 3′ to 5′: a uniform homolog sequence to amplify the plasmid and a homolog sequence to direct inserting the fragments to the genomic loci of interest. The fragments were transformed into the yeasts and integrated to the genome by homologous recombination. The integrations were confirmed by genomic DNA PCR (Yeast DNA Extraction Kit from Thermo Fisher). In the cloning of the reporter strains, we used a strain that was derived from W303 and has one-copied genomic insertion of MYOpr-MS2CP-2XGFP

and an endogenous fusion Dcp2-mRFP (*Zid and O'Shea, 2014*), as the background strain. Further, we transformed the linearized MS2-loop-containing reporter plasmids into the strain by restriction digest and genomic integration. RT-qPCR was performed to verify the one-copied genomic integration. To generate the MS2-loop-containing reporter plasmids (e.g., ZP207 pRS305-HSP30prUTR-nLuc-PEST-12XMS2-tADH1), we started from the plasmid ZP15 pRS305-12XMS2-tAdh1 (*Zid and O'Shea, 2014*). ZP15 was linearized by the restriction enzymes SacII and NotI (NEB). Promoter fragments, nanoluciferase-pest CDS fragments were inserted into linearized ZP15 using Gibson Assembly. Promoter sequences were amplified by PCR from the W303 genomic DNA. Nanoluciferase-pest CDS was amplified by PCR from the geneblock (*Masser et al., 2016*). To generate the PP7-MS2-containing reporter plasmids (e.g., ZP296 pRS305-HSP30prUTR-nLuc-PEST-1XPP7-12XMS2-tADH1), ZO680 and ZO679 were firstly annealed using the primer annealing protocol described by Thermo Fisher. ZP15 was linearized by restriction enzymes BamHI and NotI. Then annealed oligos were inserted into linearized ZP15 by T4 ligation to generate ZP440. ZP440 was further linearized by restriction enzymes SacII and NotI. Promoter fragments, nanoluciferase-pest CDS fragments were inserted into linearized ZP440 using Gibson Assembly. In the cloning of the CoTrIP experiments, detailed procedures are described in 'CoTrIP and CoTrIP analysis.'.

## Key resources table

| Reagent type (species) or resource | Designation | Source or reference | Identifiers | Additional information |
|---|---|---|---|---|
| Strain, strain background (*Saccharomyces cerevisiae*) | EY0690/w303 | Lab stock | MATa trp1 leu2 ura3 his3 can1 GAL+psi + | |
| Genetic reagent (*S. cerevisiae*) | ZY1 | This study | EY0690; pRS406-CMV lacIA-FLAG | Available in the Zid lab |
| Genetic reagent (*S. cerevisiae*) | ZY2 | This study | EY0690; HSP30-CFP-CoTrIP | |
| Genetic reagent (*S. cerevisiae*) | ZY3 | This study | ZY1; HSP30prUTR-CFP-CoTrIP | |
| Genetic reagent (*S. cerevisiae*) | ZY4 | This study | ZY1; HXK1prUTR-CFP-CoTrIP | |
| Genetic reagent (*S. cerevisiae*) | ZY5 | This study | ZY1; HSP26prUTR-CFP-CoTrIP | |
| Genetic reagent (*S. cerevisiae*) | ZY6 | This study | ZY1; GLC3prUTR-CFP-CoTrIP | (include dilution) |
| Genetic reagent (*S. cerevisiae*) | ZY7 | This study | ZY1; Blank-CoTrIP | |
| Genetic reagent (*S. cerevisiae*) | ZY18 | *Zid and O'Shea, 2014* | EY0690; MYO2pr-MS2-CP-GFP2x; Dcp2-RFP | |
| Genetic reagent (*S. cerevisiae*) | ZY147 | This study | EY0690; Rvb1-TAP | |
| Genetic reagent (*S. cerevisiae*) | ZY148 | This study | EY0690; Rvb2-TAP | |
| Genetic reagent (*S. cerevisiae*) | ZY282 | This study | EY0690; Dcp2-GFP; Rvb1-mRuby2 | |
| Genetic reagent (*S. cerevisiae*) | ZY284 | This study | EY0690; Dcp2-GFP; Rvb2-mRuby2 | |
| Genetic reagent (*S. cerevisiae*) | ZY266 | This study | ZY18; HSP30prUTR-nLuc-pest-1XPP7-12XMS2-tADH1 | |
| Genetic reagent (*S. cerevisiae*) | ZY269 | This study | ZY18, PDhh1-GFP-6xHis-PP7CP;HSP30prUTR-nLuc-pest-1XPP7-12XMS2-tADH1 | |

*Continued on next page*

*Continued*

| Reagent type (species) or resource | Designation | Source or reference | Identifiers | Additional information |
|---|---|---|---|---|
| Genetic reagent (*S. cerevisiae*) | ZY314 | This study | ZY18; Rvb1-PP7CP-6Xhis; HSP30prUTR-nLuc-pest-1XPP7-12XMS2-tADH1 | |
| Genetic reagent (*S. cerevisiae*) | ZY315 | This study | ZY18; Rvb2-PP7CP-6Xhis; HSP30prUTR-nLuc-pest-1XPP7-12XMS2-tADH1 | |
| Genetic reagent (*S. cerevisiae*) | ZY193 | This study | ZY18; HSP30prUTR-nLuc-pest-12XMS2-tADH1 | |
| Genetic reagent (*S. cerevisiae*) | ZY446 | This study | ZY18; Rvb1-PP7CP-6Xhis; HSP30prUTR-nLuc-pest-12XMS2-tADH1 | |
| Genetic reagent (*S. cerevisiae*) | ZY449 | This study | ZY18; Rvb2-PP7CP-6Xhis; HSP30prUTR-nLuc-pest-12XMS2-tADH1 | |
| Genetic reagent (*S. cerevisiae*) | ZY316 | This study | ZY18; Rvb1-PP7CP-6Xhis | |
| Genetic reagent (*S. cerevisiae*) | ZY317 | This study | ZY18; Rvb2-PP7CP-6Xhis | |
| Genetic reagent (*S. cerevisiae*) | ZY318 | This study | EY0690; Rvb1-mNeogreen | |
| Genetic reagent (*S. cerevisiae*) | ZY319 | This study | EY0690; Rvb2-mNeogreen | |
| Genetic reagent (*S. cerevisiae*) | ZY488 | This study | ZY18; HSP26prUTR-nLuc-pest-12XMS2-tADH1 | |
| Genetic reagent (*S. cerevisiae*) | ZY489 | This study | ZY18; HSP12prUTR-nLuc-pest-12XMS2-tADH1 | |
| Genetic reagent (*S. cerevisiae*) | ZY490 | This study | ZY18; HSP26prUTR-nLuc-pest-1XPP7-12XMS2-tADH1 | |
| Genetic reagent (*S. cerevisiae*) | ZY491 | This study | ZY18; HSP12prUTR-nLuc-pest-1XPP7-12XMS2-tADH1 | |
| Genetic reagent (*S. cerevisiae*) | ZY492 | This study | ZY18; HSP26prUTR-nLuc-pest-12XMS2-tADH1; Rvb1-PP7CP-6Xhis | |
| Genetic reagent (*S. cerevisiae*) | ZY493 | This study | ZY18; HSP12prUTR-nLuc-pest-12XMS2-tADH1; Rvb1-PP7CP-6Xhis | |
| Genetic reagent (*S. cerevisiae*) | ZY494 | This study | ZY18; HSP26prUTR-nLuc-pest-1XPP7-12XMS2-tADH1; Rvb1-PP7CP-6Xhis | |
| Genetic reagent (*S. cerevisiae*) | ZY495 | This study | ZY18; HSP12prUTR-nLuc-pest-1XPP7-12XMS2-tADH1; Rvb1-PP7CP-6Xhis | |

*Continued on next page*

*Continued*

| Reagent type (species) or resource | Designation | Source or reference | Identifiers | Additional information |
|---|---|---|---|---|
| Genetic reagent (*S. cerevisiae*) | ZY496 | This study | ZY18; HSP26prUTR-nLuc-pest-12XMS2-tADH1; Rvb2-PP7CP-6Xhis | |
| Genetic reagent (*S. cerevisiae*) | ZY497 | This study | ZY18; HSP12prUTR-nLuc-pest-12XMS2-tADH1; Rvb2-PP7CP-6Xhis | |
| Genetic reagent (*S. cerevisiae*) | ZY498 | This study | ZY18; HSP26prUTR-nLuc-pest-1XPP7-12XMS2-tADH1; Rvb2-PP7CP-6Xhis | |
| Genetic reagent (*S. cerevisiae*) | ZY499 | This study | ZY18; HSP12prUTR-nLuc-pest-1XPP7-12XMS2-tADH1; Rvb2-PP7CP-6Xhis | |
| Genetic reagent (*S. cerevisiae*) | ZY642 | This study | EY0690; HSP26prUTR-CFP-12XMS2-tADH1; Rvb1-TAP | |
| Genetic reagent (*S. cerevisiae*) | ZY643 | This study | EY0690; GLC3prHSP26UTR-CFP-12XMS2-tADH1; Rvb1-TAP | |
| Genetic reagent (*S. cerevisiae*) | ZY644 | This study | EY0690; HSP26prUTR-CFP-12XMS2-tADH1; Rvb2-TAP | |
| Genetic reagent (*S. cerevisiae*) | ZY645 | This study | EY0690; GLC3prHSP26UTR-CFP-12XMS2-tADH1; Rvb2-TAP | |
| Genetic reagent (*S. cerevisiae*) | ZY831 | This study | EY0690,GSY1p-GSY1ORF-pKT-ERVB-nLucPEST; RVB2-gRNA; dCas9-Mxi | |
| Genetic reagent (*S. cerevisiae*) | ZY833 | This study | EY0690,HXK1p-HXK1ORF-pKT-ERVB-nLucPEST; RVB2-gRNA; dCas9-Mxi | |
| Genetic reagent (*S. cerevisiae*) | ZY834 | This study | EY0690,HSP30p-HSP30ORF-pKT-ERVB-nLucPEST; RVB2-gRNA; dCas9-Mxi | |
| Genetic reagent (*S. cerevisiae*) | ZY362 | This study | EY0690, HSP30-pKT-ERBV1-nLucPEST | |
| Genetic reagent (*S. cerevisiae*) | ZY407 | This study | EY0690, HSP26-pKT-ERBV1-nLucPEST | |
| Genetic reagent (*S. cerevisiae*) | ZY408 | This study | EY0690, HXK1-pKT-ERBV1-nLucPEST | |
| Genetic reagent (*S. cerevisiae*) | ZY409 | This study | EY0690, GSY1-pKT-ERBV1-nLucPEST | |
| Recombinant DNA reagent | ZP66 | This study | pUC-TalO8 (Blank-CoTrIP) | Addgene:178303 |
| Recombinant DNA reagent | ZP67 | This study | TalO8-HSP30-CFP | Addgene:178304 |
| Recombinant DNA reagent | ZP68 | This study | TalO8-HXK1-CFP | |

*Continued on next page*

*Continued*

| Reagent type (species) or resource | Designation | Source or reference | Identifiers | Additional information |
|---|---|---|---|---|
| Recombinant DNA reagent | ZP69 | This study | TalO8-HSP26-CFP | Addgene:178306 |
| Recombinant DNA reagent | ZP70 | This study | TalO8-GLC3-CFP | Addgene:178307 |
| Recombinant DNA reagent | ZP64 | *Unnikrishnan et al., 2010* | pRS406-CMV-LacI-3xFLAG | Addgene:83410 |
| Recombinant DNA reagent | ZP60 | Lab stock | pFA6-TAP(CBP-TEV-ZZ)-Kan | |
| Recombinant DNA reagent | ZP61 | Lab stock | pFA6-TAP(CBP-TEV-ZZ)-His | |
| Recombinant DNA reagent | ZP47 | Lab stock | pKT-mNeongreen-Ura | |
| Recombinant DNA reagent | ZP224 | Lab stock | pFA6a-link-yoEGFP-SpHis5 | Addgene:44836 |
| Recombinant DNA reagent | ZP109 | Lab stock | pKT-mRuby2-HPH | |
| Recombinant DNA reagent | ZP93 | Carroll et al JCB 2011 | pRS316 PDhh1-GFP-6×His-PP7CP | |
| Recombinant DNA reagent | ZP296 | This study | pRS305-HSP30prUTR-nLuc-pest-1XPP7-12XMS2-tADH1 | |
| Recombinant DNA reagent | ZY311 | Lab stock | pKT-PP7CP-6xHis-tADH1 | |
| Recombinant DNA reagent | ZP207 | This study | pRS305-HSP30prUTR-nLuc-PEST-12XMS2-tADH1 | |
| Recombinant DNA reagent | ZP214 | This study | pRS305-GLC3prUTR-nLuc-PEST-12XMS2-tADH1 | |
| Recombinant DNA reagent | ZP315 | This study | pRS305-GSY1prUTR-nLuc-PEST-12XMS2-tADH1 | |
| Recombinant DNA reagent | ZP441 | This study | pRS305-HSP26prUTR-nLuc-pest-12XMS2-tADH1 | |
| Recombinant DNA reagent | ZP442 | This study | pRS305-HSP12prUTR-nLuc-pest-12XMS2-tADH1 | |
| Recombinant DNA reagent | ZP443 | This study | pRS305-HSP26prUTR-nLuc-pest-1XPP7-12XMS2-tADH1 | |
| Recombinant DNA reagent | ZP444 | This study | pRS305-HSP12prUTR-nLuc-pest-1XPP7-12XMS2-tADH1 | |
| Recombinant DNA reagent | ZP440 | Lab stock | pRS305-1XPP7-12XMS2-tADH1 | |
| Recombinant DNA reagent | ZP29 | *Zid and O'Shea, 2014* | pRS305-HSP26prUTR-CFP-12XMS2-tADH1 | |
| Recombinant DNA reagent | ZP32 | *Zid and O'Shea, 2014* | pRS305-GLC3prHSP26UTR-CFP-12XMS2-tADH1 | |
| Recombinant DNA reagent | ZP15 | *Zid and O'Shea, 2014* | pRS305-12xMS2-tADH1 | |

*Continued on next page*

*Continued*

| Reagent type (species) or resource | Designation | Source or reference | Identifiers | Additional information |
|---|---|---|---|---|
| Recombinant DNA reagent | ZP480 | *McGlincy et al., 2021* | pNTI647 dCas9-Mxi1 TetR KanMX | Addgene:139474 |
| Recombinant DNA reagent | ZP479 | *McGlincy et al., 2021* | pNTI661 pRPR1(TetO)-sgRNA | Addgene:139475 |
| Recombinant DNA reagent | ZP577 | This study | pNTI661 pRPR1(TetO)-sgRNA (Rvb2gRNA_i04) | |
| Recombinant DNA reagent | ZP345 | *Guzikowski et al., 2022* | pKT ERBV-1 nLucPEST | |
| Sequence-based reagent | NLuc+PestR | This paper | Amplify nLuc-pest to assemble into reporter vector | ATCCACTAGTTCTAGAGC TTAAACATTAATACGAGCAGAAG |
| Sequence-based reagent | yNLucF | This paper | Amplify nLuc-pest to assemble into reporter vector | ATGGTTTTTACTTTAGAAGATTTTG |
| Sequence-based reagent | HSP30pr-F | This paper | Amplify HSP30 promoter and UTR to assemble into reporter vector | TCACTATAGGGCGAATTGGAGCTCCACCGC CCTTTCTTCAAAAGTAGAAA ACTTG |
| Sequence-based reagent | HSP30utr-R | This paper | Amplify HSP30 promoter and UTR to assemble into reporter vector | TCTAAAGTAAAAACCAT TTGAAATTTGTTGTTTTTAGTAATCAA |
| Sequence-based reagent | cRvb2-R | This paper | Checking the C-terminal fusion of Rvb2 | CACCAACCAAGGCTTTTTGT |
| Sequence-based reagent | cRvb2-F | This paper | Checking the C-terminal fusion of Rvb2 | TGACCAAAACAGGTGTGGAA |
| Sequence-based reagent | cRvb1-R | This paper | Checking the C-terminal fusion of Rvb1 | CACAGCCATTACCACACCAG |
| Sequence-based reagent | cRvb1-F | This paper | Checking the C-terminal fusion of Rvb1 | CCTGAAGACGCAGAGAATCC |
| Sequence-based reagent | RVB1TAPtag_F | This paper | C-terminal TAP-tag | AAGGTCAACAAAGATTTTAGAAACTTCCGCAAATTATTTG cggatccccgggttaattaa |
| Sequence-based reagent | RVB1Taptag_R | This paper | C-terminal TAP-tag | TATTTTTATTTATGAAATGTGCTTTAGGCTTTCTTCACTG gaattcgagctcgtttaaac |
| Sequence-based reagent | RVB2TAPtag_F | This paper | C-terminal TAP-tag | TGCTAAATCAGCAGACCCTGATGCCATGGATA CTACGGAAcggatccccgggttaattaa |
| Sequence-based reagent | RVB2TAPtag_R | This paper | C-terminal TAP-tag | TATATATTTGATGCAATTTCTGCCTTAAAGTACAAAATGCgaattcgagctcgtttaaac |
| Sequence-based reagent | pKT_Rvb2_R | This paper | C-terminal tagging of pKT vector | TATATATTTGATGCAATTTCTGCCTTAAAGTACAAAATGC tcgatgaattcgagctcg |
| Sequence-based reagent | pKT_Rvb2_F | This paper | C-terminal tagging of pKT vector | TGCTAAATCAGCAGACCCTGATGCCATGGATACTACGGAA ggtgacggtgctggttta |
| Sequence-based reagent | pKT_Rvb1_R | This paper | C-terminal tagging of pKT vector | TATTTTTATTTATGAAATGTGCTTTAGGCTTTCTTCACTG tcgatgaattcgagctcg |
| Sequence-based reagent | pKT_Rvb1_F | This paper | C-terminal tagging of pKT vector | AAGGTCAACAAAGATTTTAGAAACTTCCGCAAATTATTTG ggtgacggtgctggttta |
| Sequence-based reagent | PP7_RE2 | This paper | PP7 stem loop with NotI/BamHI overhangs | GATCC TAAGGGTTTCCATATAAACTCCTTAA GC |
| Sequence-based reagent | PP7_RE1 | This paper | PP7 stem loop with NotI/BamHI overhangs | GGCCGC TTAAGGAGTTTATATGGAAACCCTTA G |
| Sequence-based reagent | Rvb2gRNA_i04rc | This paper | Reverse complement gRNA cloning oligo | gctatttctagctctaaaacGTGTGAATGTACAGTCTTCAtgccaatcgcagctcccaga |
| Sequence-based reagent | Rvb2gRNA_i04 | This paper | RVB2 gRNA cloning oligo | tctgggagctgcgattggcaTGAAGACTGTACATTCACACgttttagagctagaaatagc |

*Continued on next page*

*Continued*

| Reagent type (species) or resource | Designation | Source or reference | Identifiers | Additional information |
|---|---|---|---|---|
| Sequence-based reagent | Rvb1gRNA_i05rc | This paper | Reverse complement gRNA cloning oligo | gctatttctagctctaaaacTCTCTTCTTCATCACCACGAtgccaatcgcagctcccaga |
| Sequence-based reagent | Rvb1gRNA_i05 | This paper | RVB1 gRNA cloning oligo | tctgggagctgcgattggcaTCGTGGTGATGAAGAAGAGAgttttagagctagaaatagc |
| Sequence-based reagent | NM637 | This paper | Primers to extend gRNA oligos for Gibson Assembly ZP479 | gccttattttaacttgctatttctagctctaaaac |
| Sequence-based reagent | NM636 | This paper | Primers to extend gRNA oligos for Gibson Assembly ZP479 | ggctgggaacgaaac tctgggagctgcgattggca |
| Sequence-based reagent | Gsy1_pKTR | This paper | C-terminal tagging of pKT vector | GCTAAAAGAGTAAGATATGTTAGCAGAAGTTAAGATGGTT tcgatgaattcgagctcg |
| Sequence-based reagent | Gsy1_pKTF | This paper | C-terminal tagging of pKT vector | CGATGATGACAACGATACGTCTGCATACTACG AGGATAATggtgacggtgctggttta |
| Sequence-based reagent | HXK1_pKTR | This paper | C-terminal tagging of pKT vector | CATTACATTTTTTTCATTAAGCGCCAATGATACCAAGAGAC tcgatgaattcgagctcg |
| Sequence-based reagent | HXK1_pKTF | This paper | C-terminal tagging of pKT vector | CTGTTATTGCTGCATTGTCCGAAAAAAGAATTGCCGAAGG ggtgacggtgctggttta |
| Sequence-based reagent | HSP26_pKTR | This paper | C-terminal tagging of pKT vector | GGTCCTCGCGAGAGGGACAACACTATAGAGCC AGGTCACTtcgatgaattcgagctcg |
| Sequence-based reagent | HSP26_pKTF | This paper | C-terminal tagging of pKT vector | CAAGAAGATTGAGGTTTCTTCTCAAGAATCGTGGGGTAAC ggtgacggtgctggttta |
| Sequence-based reagent | HSP30_pKTR | This paper | C-terminal tagging of pKT vector | TGTGTTAAGCAAAGAATGATTAAGACAATCTC AAGCTGCTtcgatgaattcgagctcg |
| Sequence-based reagent | HSP30_pKTF | This paper | C-terminal tagging of pKT vector | ACCCGAACCTGAAGCAGAGCAAGCTGTCGAAG ATACTGCTggtgacggtgctggttta |
| Sequence-based reagent | qMS2-CP-R | This study | qPCR primers | GTCGGAATTCGTAGCGAAAA |
| Sequence-based reagent | qMS2-CP-F | This study | qPCR primers | GCAGAATCGCAAATACACCA |
| Sequence-based reagent | qnLuc_R | This study | qPCR primers | CCTTCATAAGGACGACCAAA |
| Sequence-based reagent | qnLuc_F | This study | qPCR primers | TGGTGATCAAATGGGTCAAA |
| Sequence-based reagent | qHsp12prR | This study | qPCR primers | GAGCGGGTAACAGATGGAAG |
| Sequence-based reagent | qHsp12prF | This study | qPCR primers | GCGCTGCAAGTTCCTTACTT |
| Sequence-based reagent | qHsp104prR | This study | qPCR primers | ATGAAACTCTCGCCACAACC |
| Sequence-based reagent | qHsp104prF | This study | qPCR primers | AAATGGACTGGATCGACGAC |
| Sequence-based reagent | qHsp26R | This study | qPCR primers | ATCATAAAGAGCGCCAGCAT |
| Sequence-based reagent | qHsp26F | This study | qPCR primers | AACAGATTGCTGGGTGAAGG |
| Sequence-based reagent | qGsy1prR | This study | qPCR primers | GCGGGAAGAAAAGAAGGAGT |
| Sequence-based reagent | qGsy1prF | This study | qPCR primers | AGGGCAGACAAGAGGCTGTA |

*Continued on next page*

*Continued*

| Reagent type (species) or resource | Designation | Source or reference | Identifiers | Additional information |
|---|---|---|---|---|
| Sequence-based reagent | qActR | *Zid and O'Shea, 2014* | qPCR primers | CGGTGATTTCCTTTTGCATT |
| Sequence-based reagent | qActF | *Zid and O'Shea, 2014* | qPCR primers | CTGCCGGTATTGACCAAACT |
| Sequence-based reagent | qTub1prR | This study | qPCR primers | CGCTAGATGCATTAAACATGAAG |
| Sequence-based reagent | qTub1prF | This study | qPCR primers | GTGCTCACACCAAGCATCAT |
| Sequence-based reagent | qAct1prR | This study | qPCR primers | GAGAGGCGAGTTTGGTTTCA |
| Sequence-based reagent | qAct1prF | This study | qPCR primers | TCACCCGGCCTCTATTTTC |
| Sequence-based reagent | qGPH1prR | This study | qPCR primers | TCGTCGGTGTTCCTTCCTTA |
| Sequence-based reagent | qGPH1prF | This study | qPCR primers | GAACGCCTTCCCCAATTAC |
| Sequence-based reagent | qHxk1prR | This study | qPCR primers | CCTGGTTGCTCCAGTAAGG |
| Sequence-based reagent | qHxk1prF | This study | qPCR primers | TTCAGGAAGAATGGCAGTCC |
| Sequence-based reagent | qGlc3prR | This study | qPCR primers | TTGCAACAGCCCCTTGGAC |
| Sequence-based reagent | qGlc3prF | This study | qPCR primers | GGGCACTCATCAACAATGTG |
| Sequence-based reagent | qHsp26prF | This study | qPCR primers | CTGTCAAGGTGCATTGTTGG |
| Sequence-based reagent | qHsp30prR | This study | qPCR primers | CGGGATATGGCTTTGCTTAC |
| Sequence-based reagent | qHsp30prF | This study | qPCR primers | CGATTTTGTTGGCCATTTTCCA |
| Sequence-based reagent | qGsy1R | *Zid and O'Shea, 2014* | qPCR primers | GCAGTGATTTGCGACACAGT |
| Sequence-based reagent | qGsy1F | *Zid and O'Shea, 2014* | qPCR primers | GCCGCTGGTGATGTAGATTT |
| Sequence-based reagent | qHsp12R | *Zid and O'Shea, 2014* | qPCR primers | TTGGTTGGGTCTTCTTCACC |
| Sequence-based reagent | qHsp12F | *Zid and O'Shea, 2014* | qPCR primers | CGAAAAAGGCAAGGATAACG |
| Sequence-based reagent | qHsp104R | *Zid and O'Shea, 2014* | qPCR primers | CACTTGGTTCAGCGACTTCA |
| Sequence-based reagent | qHsp104F | *Zid and O'Shea, 2014* | qPCR primers | CGACGCTGCTAACATCTTGA |
| Sequence-based reagent | qGph1R | *Zid and O'Shea, 2014* | qPCR primers | TCATAAGCAGCCATGTCATCA |
| Sequence-based reagent | qGph1F | *Zid and O'Shea, 2014* | qPCR primers | TTCCCCAAGAAATCAAGTCAA |
| Sequence-based reagent | qTub1R | *Zid and O'Shea, 2014* | qPCR primers | GGTGTAATGGCCTCTTGCAT |
| Sequence-based reagent | qTub1F | *Zid and O'Shea, 2014* | qPCR primers | CCACGTTTTTCCATGAAACC |

*Continued on next page*

*Continued*

| Reagent type (species) or resource | Designation | Source or reference | Identifiers | Additional information |
|---|---|---|---|---|
| Sequence-based reagent | qHsp30R | *Zid and O'Shea, 2014* | qPCR primers | TCAGCTTGAACACCAGTCCA |
| Sequence-based reagent | qHsp30F | *Zid and O'Shea, 2014* | qPCR primers | GGGCAGTGTTTGCAGTCTTT |
| Sequence-based reagent | qGlc3R | *Zid and O'Shea, 2014* | qPCR primers | CGAAATCGCCGTTAGGTAAA |
| Sequence-based reagent | qGlc3F | *Zid and O'Shea, 2014* | qPCR primers | CAATCCGGAAACCAAAGAAA |
| Sequence-based reagent | qHsp26prF | *Zid and O'Shea, 2014* | qPCR primers | CATAAGGGGGAGGGAATAAC |
| Sequence-based reagent | qCITCFPf | *Zid and O'Shea, 2014* | qPCR primers | CTGGTGAAGGTGAAGGTGAC |
| Sequence-based reagent | qCITCFPr | *Zid and O'Shea, 2014* | qPCR primers | TGTGGTCTGGGTATCTAGCG |
| Sequence-based reagent | qHXK1F | *Zid and O'Shea, 2014* | qPCR primers | TTTGTAGCAATGGGACGACA |
| Sequence-based reagent | qHXK1R | *Zid and O'Shea, 2014* | qPCR primers | GTACCCAGCTTCCCAAAACA |

## Yeast growth and media

The background yeast strain w303 (EY0690) was used for all experiments. For cells cultured in the functional experiments, cells were streaked out on the yeast extract peptone dextrose (YPD) agarose plate (BD) from the frozen stocks and grew at 30°C for 2 days. Single colony was selected to start the overnight culture for each biological replicate. Cells were grown at 30°C in batch culture with shaking at 200 r.p.m. in synthetic complete glucose medium (SCD medium: yeast nitrogen base from RPI, glucose from Sigma-Aldrich, SC from Sunrise Science). When the $OD_{660}$ of cells reached 0.4, half of the culture was harvested as the pre-starved sample. The other half of the culture was transferred to the prewarmed synthetic complete medium lacking glucose (SC-G medium) by centrifugation method. Cells are centrifuged at 3000 × $g$, washed once by SC medium, and resuspended in the same volume as the pre-starvation medium of SC medium. Glucose starvation was performed at the same 200 r.p.m. shaking speed and 30°C. The length of the glucose starvation time varies from 10 min to 30 min depending on the experiments.

## CoTrIP and CoTrIP analysis

The protocol was developed based on *Unnikrishnan et al., 2012*. ZP64 PRS406-CMV-lacI-FLAG was integrated into the W303 yeast background by linearization within the *URA3* gene with BstBI digestion and transformation into yeast. The CoTrIP plasmid was constructed by modifying the ZP66 pUC-TALO8 plasmid that contains eight copies of the Lac operator. The EcoRI sites in ZP66 were mutated to NotI using QuikChange II Site-directed mutagenesis kit (Stratagene). Promoter-specific reporters driving CFP were inserted into pUC-TALO8-NotI by digesting with NheI followed by Gibson Assembly. The plasmid backbone was then digested with NotI, gel purified, ligated, and transformed into yeast using standard lithium acetate transformation. Then, 1 L of yeast were grown overnight in SCD medium-Trp to maintain selection on the CoTrIP plasmid, until an $OD_{660}$ 0.3–0.4. Cells were filtered, washed with SC-G -Trp media, and resuspended in 1 L of prewarmed media and grown at 30°C for 5 min. Cells were then refiltered, resuspended in 4 mL of PBS in a glass Petri dish, and crosslinked using UV from a Stratalinker 1800 (254 nm, 9999 microjoules × 100, 5 cm from the UV bulb). Crosslinked cells were pelleted and resuspended in 2 mL of Buffer H 150 (25 mM HEPES KOH pH 7.6, 2 mM $MgCl_2$, 0.5 mM EGTA, 0.1 mM EDTA, 10% glycerol, 150 mM KCl, 0.02% NP40) plus protease inhibitor (P8215 MilliporeSigma) and then dripped into liquid $N_2$ to be cryogenically ball milled using a Retsch PM100. Ground lysate was clarified by spinning at 3500 × $g$ for 5 min at 4°C, isolating the supernatant and spinning at 12K for 5 min at 4°C. Supernatant was aliquoted and frozen at –80°C. Then, 10 μL of unpacked Anti-FLAG M2 Magnetic Beads (M8823 MilliporeSigma) per sample were pre-washed with

Buffer H 150. Then, 300 µL of extract was added to magnetic beads and incubated at 4°C for 3 hr with rotation. Beads were then washed three times with Buffer H 150, three times with Buffer H 300 (300 mM KCl), and once with Buffer H 150. Then, 500 µg/mL 3xFLAG peptide (F4799 MilliporeSigma) was diluted in Buffer H 150 and CoTrIP plasmids were eluted with 100 µL elution buffer with FLAG peptide. Elutions were taken forward for DNA, RNA, and mass spectrometry. Mass spectrometry was performed through the Yeast Resource Center by James Moresco of the Yates lab and was funded through a P41GM103533 Biomedical Technology Resource Center grant. Data-dependent acquisition of MS/MS spectra was performed with an LTQ-Orbitrap. Tandem mass spectra were extracted from raw files using RawExtract 1.9.9 (*McDonald et al., 2004*) and were searched against a yeast protein database (http://www.yeastgenome.org) using ProLuCID (*Peng et al., 2003*; *Xu et al., 2015*). Enrichment analysis was performed using the CRAPome online data analysis software (https://reprint-apms.org/; *Mellacheruvu et al., 2013*).

## ChIP-sequencing

The protocol was developed based on *Grably and Engelberg, 2010*. Pgk1-TAP was included as a negative control that should not be bound to chromatin to control for previous effects showing that highly expressed regions are hyper-ChIPable in yeast even with no tag present (*Teytelman et al., 2013*). Then, 100 mL of yeast were grown overnight in SCD medium, until an $OD_{660}$ around 0.4. Also, 50 mL of cells were filtered, washed with SC-G media, and resuspended in 50 mL of prewarmed media and grown at 30°C for 10 min. Then, 50 mL of pre-starved and 50 mL of 10 min glucose-starved cell culture was fixed by incubating in the freshly made crosslink buffer (1% formaldehyde, 10 mM NaCl, 0.1 mM EDTA, 5 mM HEPES pH 7.5), respectively, with gentle shaking at room temperature for 15 min. Crosslink was quenched by introducing 0.5 M of glycine for 5 min at room temperature. Cells were harvested by centrifugation at 3000 × *g* at 4°C, washed twice in the ice-cold TBS buffer (20 mM Tris pH 7.5, 150 mM NaCl). Cells were resuspended in 400 µL of ChIP lysis buffer (50 mM HEPES-KOH pH 7.5, 150 mM NaCl, 1 mM EDTA, 1% Triton X-100, 0.1% sodium deoxycholate, 1 mM PMSF, 0.5% SDS), and lysed by bead-beating (Biospec Products) for 1 min five times. Lysis was verified under microscope. Lysates were sonicated by Covaris Sonicator to ~500 bp fragments (130 µL/tube, 105 PIP, 5% Duty F, 200 cycles/burst, 80 s). Lysates were centrifuged at 15,000 × *g* at 4°C to remove the cell debris and diluted to 1 mL. Save 10% of the clear lysate to verify the sonication by protein digestion using Pronase, reverse crosslinking, RNA digestion using RNase A, and running the samples on 1% agarose gel. Then, 50 µL of IgG-Dynabeads per sample was used. The protocol of making the IgG-Dynabeads from Dynabeads M-270 Epoxy (Thermo Fisher) was taken from *Li, 2011*. IgG-Dynabeads were pre-washed three times with ChIP lysis buffer. And 1% of the lysate was saved as the input and for Western blotting, respectively. The IP sample was incubated with IgG-Dynabeads, rotating at 4°C for 4 hr. The IP samples were further washed twice by ChIP lysis buffer with 0.1% SDS, twice by ChIP lysis buffer with 0.1% SDS and 0.5 M NaCl, once by ChIP wash buffer (50 mM Tris pH 7.5, 0.25 mM LiCl, 1 mM EDTA, 0.5% NP-40, 0.5% sodium deoxycholate), and once by TE buffer pH 7.5. Before the last wash, save 10% of the sample for Western blotting. Western blotting was performed to verify the successful enrichment and purification of the protein of interest. Samples were later eluted from beads in 250 µL 2X Pronase buffer (50 mM Tris pH 7.5, 10 mM EDTA, 1% SDS) at 65°C for 10 min and beads were removed. Samples were then digested by 1.6 mg/mL Pronase (20 µL of 20 mg/mL Pronase, Sigma-Aldrich) at 42°C for 2 hr and reverse-crosslinked at 65°C for 6 hr. DNA was extracted from the sample using phenol-chloroform, washed once with chloroform, and pelleted using ethanol, washed twice with 75% ethanol (Sigma-Aldrich), and resuspended in 200 µL TE buffer. DNA sequencing libraries were prepared and sequenced following the NextSeq 500 SR 75 protocol at The UCSD IGM Genomics Center (Illumina).

## ChIP-sequencing analysis

Sequencing reads were demultiplexed and the quality was verified by FastQC (*Anders, 2010*). The reads were mapped to S288C *Saccharomyces cerevisiae* genome using Bowtie2 (*Langmead and Salzberg, 2012*). The mapped reads were then sorted, indexed, and converted into BAM files using SAMtools (*Li et al., 2009*). Duplicated reads were removed using Picard Tools (*Broad Institute, 2009*). The genome browser files for visualizing the reads were generated by igvtools and were visualized on Integrative Genomics Viewer by Broad Institute and Sushi.R (*Phanstiel et al., 2014*; *Robinson*

*et al., 2011*; *Thorvaldsdóttir et al., 2013*). Enriched peaks for Rvb1/Rvb2 were called using MACS (*Zhang et al., 2008*). deepTools was used to calculate and plot the enrichment of Rvb1/Rvb2 on genome and selected genes (*Ramírez et al., 2016*). The coverage of reads (BigWig files) was calculated from indexed BAM files using bamCoverage. To compare the enrichment of two targets, the matrix was computed from BigWig files of two targets using computeMatrix and further presented as a heatmap by plotHeatmap. To visualize Rvb1/Rvb2's enrichment, the matrix was computed from BigWig files of the target and S288C genome annotation as the reference using computeMatrix and further presented by plotProfile. Additionally, BEDtools was used to calculate the coverage of regions of interest and a home-made R script was generated to analyze and plot the enrichment of Rvb1/Rvb2 on genome and specific gene groups (*Quinlan and Hall, 2010*).

## RNA immunoprecipitation (RIP)

The RIP protocol was developed based on the protocol from *Van Nostrand et al., 2016*; *Zander et al., 2016*. Also, 100 mL of yeast were grown overnight in SCD medium, until an $OD_{660}$ around 0.4. Then, 50 mL of cells were filtered, washed with SC-G media, and resuspended in 50 mL of prewarmed media and grown at 30°C for 15 min. Then, 50 mL of pre-starved and 15 min glucose-starved cell culture was washed and resuspended in 10 mL of ice-cold PBS buffer, fixed by UV irradiation on a 10 cm Petri dish using a Stratalinker 1800 (254 nm, 9999 microjoules × 100, 5 cm from the UV bulb), and harvested. Cells were resuspended in 400 µL of ice-cold RIP lysis buffer (50 mM Tris pH 7.5, 100 mM NaCl, 1% NP-40, 0.5% SDS, 0.2 mM PMSF, 1 mM DTT, 10U RNase inhibitor from Promega, cOmplete Protease Inhibitor Cocktail from Roche), and lysed by bead-beating (Biospec Products) for 1 min five times. Bright-field microscopy was used to verify that more than 90% of cells were lysed. The lysates were centrifuged softly at 1000 × *g* at 4°C for 10 min to remove cell debris and diluted to 500 µL. Clear lysates were treated with 5U RQ1 DNase and 5U RNase inhibitor (Promega) at 37°C for 15 min. Then 1% of the lysate was saved as the input and for Western blotting, respectively. Also, 50 µL of IgG-Dynabeads per sample was used. The protocol of preparing the IgG-Dynabeads from Dynabeads M-270 Epoxy (Thermo Fisher) was taken from *Li, 2011*. IgG-Dynabeads were pre-washed three times with RIP lysis buffer. The IP samples were incubated with IgG-Dynabeads, rotating at 4°C for 4 hr. The IP samples were further washed six times by RIP wash buffer (50 mM Tris pH 7.5, 100 mM NaCl, 0.1% NP-40) at 4°C. Samples were later eluted from the beads in 100 µL of PK buffer (100 mM Tris pH 7.5, 50 mM NaCl, 10 mM EDTA) at 65°C for 15 min and later the proteins were digested by 10U Proteinase K (NEB) at 37°C for 30 min. Digestion was later activated by incubation with urea (210 mg/mL) at 37°C for 20 min. RNA was extracted using TRIzol reagent (Thermo Fisher) according to the vendor's protocol. RNA was washed twice by 70% EtOH and eluted in 10 µL of RNase-free water. RNA samples were further digested fully by RQ1 DNase (Promega) in 10 µL system and were reverse transcribed by ProtoScript II reverse transcriptase (NEB) (a 1:1 combination of oligo dT18 and random hexamers was used to initiate reverse transcription). The cDNA was investigated by RT-qPCR.

## Live-cell microscopy and analysis

Cells were grown to an $OD_{660}$ to ~0.4 in SCD medium at 30°C and glucose-starved in SC-G medium for 15 and 30 min. Then, 100 µL of cell culture was loaded onto a 96-well glass-bottom microplate (Cellvis). Cells were imaged using an Eclipse Ti-E microscope (Nikon) with an oil-immersion ×63 objective. Imaging was controlled using NIS-Elements software (Nikon). Imaging analysis was performed on Fiji software.

## Nanoluciferase assay and analysis

The nanoluciferase (nLuc) assay was adapted from methods previously described by *Masser et al., 2016*. Cells were grown to an $OD_{660}$ to ~0.4 in SCD medium at 30°C and glucose-starved in SC-G medium for 30 min. Then, 90 µL of cell culture was loaded onto a Cellstar non-transparent white 96-well flat-bottom plate (Sigma-Aldrich). $OD_{660}$ of cells was taken for each sample. For cells treated with cycloheximide (CHX), CHX was added to a final concentration of 100 µg/mL to stop the translation for 5 min. To measure the nanoluciferase signal, 11 µL of substrate mix (10 µL of Promega Nano-Glo Luciferase Assay Buffer, 0.1 µL of Promega NanoLuc luciferase substrate, and 1 µL of 10 mg/mL CHX) was added and mixed with the samples by pipetting. Measurements were taken immediately after addition of substrate mix by Tecan Infinite Lumi plate reader. To analyze the data, the luciferase level

of samples was firstly divided by the $OD_{660}$ level of the samples. Then the normalized luciferase level of non-CHX-treated sample was further normalized by subtracting the luciferase level of CHX-treated sample. For glucose readdition experiments, cells were starved for 30 min and then 2% glucose was added back to the cultures and then the luciferase production was followed over a 10 min period.

## RVB2 CRISPRi knockdown

RVB2 gRNA forward and reverse complement oligos were annealed together and then further extended using NM637 and NM636. This PCR product was inserted into the TetO gRNA vector pNTI661 by digesting this vector with BamHI/HindIII as described previously (*McGlincy et al., 2021*). Cells were grown overnight in SC-Leu+Glu media to low OD < 0.5. Cells were diluted and 250 ng/L of ATc was added to the experimental sample and control and experimental samples were allowed to grow for 8 hr to an OD ~0.4. Yeast were either prepped for assays or glucose-starved for 30 min and then prepped for nLuc and RT-qPCR assays.

## Western blotting

The Western blotting protocol was adapted from *Tsuboi et al., 2020*. IP and input samples were mixed with the same volume of 2X Laemmli buffer (Bio-Rad) and were boiled at 95°C for 10 min. The samples were then resolved by SDS-PAGE (Bio-Rad), and a rabbit polyclonal antibody specific for calmodulin-binding peptide (A00635-40, GenScript), a Goat anti-Rabbit IgG (H+L) Secondary Antibody, HRP (Thermo Fisher), and SuperSignal West Femto Maximum Sensitivity Substrate (Thermo Fisher) were used to detect TAP-tagged proteins. The blotting was imaged using a Gel Doc XR+ Gel Documentation System (Bio-Rad).

## Real-time quantitative PCR

The RT-qPCR protocol was adapted from *Tsuboi et al., 2020*. RNA was extracted using the Master-Pure Yeast RNA Purification Kit (Epicentre). cDNA was prepared using ProtoScript II Reverse Transcriptase (NEB #M0368X) with a 1:1 combination of oligodT 18 primers and random hexamers (NEB) according to the manufacturer's instructions. mRNA abundance was determined by qPCR using a home-brew recipe with SYBR Green at a final concentration of 0.5× (Thermo Fisher #S7564). Primers specific for each transcript are described in Key resources table. The mRNA levels were normalized to *ACT1* abundance, and the fold change between samples was calculated by a standard ΔΔCt analysis. All data were included except for one sample that had high technical variation, and another that had very high *ACT1* CT values. Both samples were flagged as analysis began.

## Mathematical modeling on the mRNA induction

The mathematical modeling method was adapted from *Elkon et al., 2010* and performed in Python Jupyter Notebook (https://jupyter.org/). To accurately describe the dynamics of induced mRNA transcription, we used an ordinary differential equation as follows:

$$\frac{dX}{dt} = \beta - \alpha X$$

where X is the mRNA concentration, α is the degradation constant, and β is the transcription rate. We assumed that transcription and degradation rates play essential roles in shaping the overall curve of mRNA increase, and these parameters stay constant over the course of induced expression. We then hypothesized that Rvb1/Rvb2 binding to mRNAs could either increase β or decrease α, leading to greater mRNA abundance than the PP7 control. To observe the effects of varied transcription or degradation rates on the mRNA abundance, we solved the differential equations with different parameters using ODEINT algorithm, and generated time profiles of the mRNA fold in log2 scale. Solution to the differential equation was expressed as the following function of change in X with respect to time:

$$\Delta X\left(t\right) = \left(\frac{\beta}{\alpha} - X_0\right)\left(1 - e^{-\alpha t}\right)$$

where $X_0$ is the mRNA level at t = 0, the initial time of mRNA induction. Since the degradation rate was proportional to the mRNA concentration, we expected the curves to have a steady increase, followed by a gradual leveling off where the mRNA concentrations stay constant over time. Closer

look at the differential equation showed that at steady state (dX/dt = 0) the mRNA concentration is determined by the ratio of β to α:

$$X_{ss} = \frac{\beta}{\alpha}$$

whereas for the time it takes for the curve to transition into steady state, inversely proportional to the degradation constant, is given by

$$T_{1/2} = \ln\left(\frac{2}{\alpha}\right)$$

Thus, we showed that for a log-log plot, the expected shape of the curves can be altered by varying α and β. At constant α, increasing β would only shift the curve up, while at constant β, increasing α would cause the mRNA abundance to enter steady state more rapidly. Through the comparison between mathematical modeling and experimental data, we could infer the actual effects of Rvb1/Rvb2 binding to mRNA on the transcription and decay rates.

### Ribosome profiling

The ribosome profiling protocol was adapted from *Zid and O'Shea, 2014*. Yeast was grown in SCD to an $OD_{660}$ between 0.3 and 0.4. Then, cells were collected by filtration, resuspended in SC-G medium. After 15 min, 2% glucose was added back. CHX was added to a final concentration of 0.1 mg/mL for 1 min, and cells were then harvested. Cells were pulverized in a PM 100 ball mill (Retsch), and extracts were digested with RNase I followed by the isolation of ribosome-protected fragments by purifying RNA from the monosome fraction of a sucrose gradient. Isolated 28-base sequences were polyadenylated, and reverse transcription was performed using OTi9pA. OTi9pA allowed samples to be multiplexed at subsequent steps. RNA-seq was performed on RNA depleted of rRNA using a yeast Ribo-Zero kit (Epicentre). Samples were multiplexed and sequenced on a HiSeq analyzer (Illumina).

To analyze the ribosomal profiling and RNA-seq sequences, reads were trimmed of the 39 run of poly(A)s and then aligned against *S. cerevisiae* rRNA sequences using Bowtie sequence aligner (*Langmead and Salzberg, 2012*). Reads that did not align to rRNA sequences were aligned against the full *S. cerevisiae* genome. Reads that had an unambiguous alignment with less than three mismatches were used in the measurements of ribosome occupancy and mRNA levels. Since there were many reads mapping to the initiation region (216 bp to 120 bp in relation to the AUG), the ribosome occupancy for each mRNA was calculated by taking the total number of ribosome reads (normalized to the total number of aligned reads in reads per million reads [RPM]) in the downstream region (120 bp from the AUG to the end of the ORF) and dividing this by the number of mRNA reads (RPM) in the same region. The ribosome occupancy along the mRNA was calculated by dividing the ribosome read counts at each base pair along the gene by the average number of mRNA reads per base pair for each gene.

## Acknowledgements

We thank the Zid lab especially Anna R Guzikowski and Tatsuhisa Tsuboi for helpful feedback on this manuscript. We also thank Toshio Tsukiyama for sharing the LacI-Flag and pUC-TalO8 plasmids. We also thank the Ingolia lab for sharing the dCas9-Mxi pNTI647 and gRNA base vector pNTI661. This work was in part supported by the National Institutes of Health R35GM128798 (to BMZ) and the Yeast Resource Center P41GM103533 (JJM and JRY).

## Additional information

### Funding

| Funder | Grant reference number | Author |
| --- | --- | --- |
| National Institute of General Medical Sciences | R35GM128798 | Brian M Zid |

| Funder | Grant reference number | Author |
|---|---|---|
| National Institute of General Medical Sciences | P41GM103533 | James J Moresco |

The funders had no role in study design, data collection and interpretation, or the decision to submit the work for publication.

## Author contributions

Yang S Chen, Conceptualization, Formal analysis, Investigation, Methodology, Writing - original draft, Writing - review and editing; Wanfu Hou, Vince Harjono, Formal analysis, Investigation, Writing - review and editing; Sharon Tracy, James J Moresco, Investigation, Methodology; Alex T Harvey, Fan Xu, Formal analysis, Investigation; John R Yates, Resources, Funding acquisition; Brian M Zid, Conceptualization, Formal analysis, Supervision, Funding acquisition, Writing - original draft, Project administration, Writing - review and editing

## Author ORCIDs

Yang S Chen (ID) http://orcid.org/0000-0003-3174-091X
Fan Xu (ID) http://orcid.org/0000-0002-0041-4276
John R Yates III, (ID) http://orcid.org/0000-0001-5267-1672
Brian M Zid (ID) http://orcid.org/0000-0003-1876-2479

## Decision letter and Author response

Decision letter https://doi.org/10.7554/eLife.76965.sa1
Author response https://doi.org/10.7554/eLife.76965.sa2

# Additional files

## Supplementary files

• MDAR checklist

## Data availability

ChIP-sequencing reads were deposited at GEO. The raw files and analyzed ChIP-seq enrichment data generated in this study is available at GEO: GSE184473. Ribosome profiling sequencing reads are deposited at GEO: GSE200491. CoTrIP plasmids can be obtained through Addgene - 178303, 178304, 178306. Further information and requests for resources and reagents should be directed to and will be fulfilled by the corresponding contact, B.M.Z. (zid@ucsd.edu).

The following dataset was generated:

| Author(s) | Year | Dataset title | Dataset URL | Database and Identifier |
|---|---|---|---|---|
| Zid BM, Chen YS | 2021 | ChIP-seq facilitates the quantitative analysis of Rvb proteins' enrichment on the genome during stress | https://www.ncbi.nlm.nih.gov/geo/query/acc.cgi?acc=GSE184473 | NCBI Gene Expression Omnibus, GSE184473 |

The following previously published datasets were used:

| Author(s) | Year | Dataset title | Dataset URL | Database and Identifier |
|---|---|---|---|---|
| Zid BM, O'Shea EK | 2014 | Ribosome profiling upon glucose starvation in *S. cerevisiae* | https://www.ncbi.nlm.nih.gov/geo/query/acc.cgi?acc=GSE56622 | NCBI Gene Expression Omnibus, GSE56622 |
| Zid BM | 2022 | Ribosome profiling and RNA-seq of an acute glucose starvation timecourse and 5 day growth course in *S. cerevisiae* | https://www.ncbi.nlm.nih.gov/geo/query/acc.cgi?acc=GSE200491 | NCBI Gene Expression Omnibus, GSE200491 |

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
