## [Editor Report]

This study provides convincing evidence that the Rvb1 and Rvb2 proteins preferentially associate with the promoters of a class of genes that exhibit increased mRNA expression but translational repression and association with mRNA granules in response to acute glucose starvation. They show that Rvb1/Rvb2 associate with the target mRNAs in addition to their promoters, in a manner requiring only the gene promoter. Tethering Rvb1 or Rvb2 to a reporter mRNA is sufficient to repress its translation, stimulate its synthesis, and induce its sequestration in cytoplasmic granules; whereas Rvb2 knockdown eliminates the translational repression of several reporter mRNAs. These compelling findings support the important demonstration that Rvb1/Rvb2 are loaded on transcripts co-transcriptionally and accompany them into the cytoplasm where they repress translation in a manner associated with the accumulation of the repressed mRNAs in granules.

---

## [Decision Letter]

[Editors' note: this paper was reviewed by Review Commons.]

---

## [Author Response]

We thank the reviewers for their expertise and thorough comments on our manuscript. We believe their comments and our revisions have strongly improved the manuscript. Point by point replies to comments are given below.

Reviewer #1 (Evidence, reproducibility and clarity (Required)):This is a very interesting paper with novel observations. The authors find that, in yeast, Rvb1/2 AAA+ ATPases couple transcription, mRNA granular localization, and mRNAs translatability during glucose starvation. Rvb1 and Rvb2 were found to be enriched at the promoters and mRNAs of genes involved in alternative glucose metabolism pathways that are transcriptionally upregulated but translationally downregulated during glucose starvation.The following are some commentsIntroduction1. "Structural studies have shown that they form a dodecamer comprised of a stacked Rvb1 hexametric ring and a Rvb2 hexametric ring."– Rvb1 and Rvb2 form a heterohexameric ring with alternating arrangement (not homohexamers that stack on top of each other as suggested by this sentence)– In yeast, they oligomerize mostly as single hexametric rings, with dodecamers reported being less than 10% in frequency in vivo (eg Jeganathan et al. 2015 https://doi.org/10.1016/j.jmb.2015.01.010)

We have changed the text to more appropriately reflect this. “Structural studies have shown that in yeast they form an alternating heterohexameric ring or two stacked heterohexameric rings (Jeganathan et al., 2015).”

ResultsSection: Rvb1/Rvb2 are identified as potential co-transcriptionally loaded protein factors on the alternative glucose metabolism genes1. "These two proteins are generally thought to act on DNA but have been found to be core components of mammalian and yeast cytoplasmic stress granules"– These two papers extensively show Rvb1/Rvb2 localization to granules/condensates under stress/nutrient starvation conditions and should be cited. The Rvb1/2 foci were named Rbits:i. Rizzolo et al. 2017 https://doi.org/10.1016/j.celrep.2017.08.074ii. Kakihara et al. 2014 https://doi.org/10.1186/s13059-014-0404-4

We have added citations to these two relevant papers.

2. "a portion of them becomes localized to cytoplasmic granules that are not P-bodies in both 15-minute and 30-minute glucose starvation conditions (Figure 1—figure supplement 2)"– Supplement figure 2 only includes results under 30-min glucose starvation, no 15-min data was shown

We have corrected the text and only reference 30-minute glucose starvation.

3. Figure 1C, unclear whether p-value here is for FC of GLC3 over HSP or FC of GLC3 over CRAPome. In addition, both FC datasets should have p-values.

We have clarified the p-value comes from our GLC3 vs HSP comparison in the figure legend. Unfortunately, we did not have the absolute spectral counts from the FC versus the CRAPome and could not calculate significance for the CRAPome comparison.

Section: Rvb1/Rvb2 are enriched at the promoters of endogenous alternative glucose metabolism genes1. "Here, we performed ChIP-seq on Rvb1, Rvb2, and the negative control Pgk1 in 10 minutes of glucose starvation (Figure 2—figure supplement 3, left)"- Unclear what figure is being referred to, panel A or panel B?

We revised to the following. “Here, we performed ChIP-seq on Rvb1, Rvb2, and the negative control Pgk1 in 10 minutes of glucose starvation (the Western validation of Rvb1 and Rvb2’s immunoprecipitation is showed as Figure 2—figure supplement 3A)”

2. "Structural studies have shown that Rvb1/Rvb2 can form a dodecamer complex. Their overlapped enrichment also indicates that Rvb1 and Rvb2 may function together."- They function together regardless of forming a dodecamer or not, as they assemble as heterohexamers

We have revised to the following. “Structural studies have shown that Rvb1/Rvb2 assemble as heterohexamers (Gribun, Cheung, Huen, Ortega, & Houry, 2008). Their overlapped CHIP enrichment further supports that Rvb1 and Rvb2 function together along DNA.”

Section: Engineered Rvb1/Rvb2 tethering to mRNAs directs the cytoplasmic localization and repressed translation1. Does binding of any protein to PP7 loop in this construct alter cytoplasmic fate? A control such as GFP-CP or any other protein attached to CP should be used.

As suggested, we have expressed GFP-PP7-CP plasmid in the HSP30prUTR-nLuc-PP7-MS2 yeast strain and found that this tethering has no significant effect on protein synthesis induction, mRNA induction, and translatability (Figure 4B).

2. No statistical analysis was done for Figure 4E quantification

We have added statistical analysis for Figure 4E (now Figure 4D).

3. "Results showed that after replenishing the glucose to the starved cells, the translation of those genes is quickly induced, with an ~8-fold increase in ribosome occupancy 5 minutes after glucose re-addition for Class II mRNAs (Figure 4—figure supplement 9)"– Would be important to see this recovery (increase in translation after glucose replenishment) in one of the reporter constructs used in the paper, such as GL3 promoter driven CFP.

Unfortunately, tagging of the endogenous GLC3 gene with nLuc disrupted normal regulation of GLC3 and it became constitutively expressed. We have instead tagged other Class II mRNAs (GSY1 and HXK1) as well as Class I mRNAs (HSP30 and HSP26) and found that the Class II reporters show more robust induction upon glucose re-addition than Class I mRNAs further supporting our ribosome profiling data (Figure 4—figure supplement 9B).

Section: Engineered Rvb1/Rvb2 binding to mRNAs increases the transcription of corresponding genes1. How many biological replicates is in Figure 5B? There does not seem to be any error bars/gray sections indicating sample variation. P-value was also not calculated.

We have added the number of biological replicates to the figure legend (n=4) and statistical analysis to Figure 5B (now Figure 4E).

Reviewer #1 (Significance (Required)):This is a very interesting manuscript that ascribes yet another function of the highly conserved RVB1/2 AAA+ ATPases.Referee Cross-commentingAll reviewers agree that this an interesting paper. However, the reviewers do suggest specific experiments to verify some of the results. Carrying out these experiments will definitely improve the paper.

Reviewer #2 (Evidence, reproducibility and clarity (Required)):

In their manuscript entitled "Rvb1/Rvb2 proteins couple transcription and translation during glucose starvation", Chen and co-authors use genetics and microscopy to demonstrate how budding yeast regulate cytoplasmic translation by their promoter sequences by two conserved ATPases Rvb1 and Rvb2 during nutrient stress. The authors show that these two ATPases repress translation of target mRNAs and then propose that these two proteins also recruit mRNAs to P bodies. The authors show that Rvb1/2 preferentially binds in the presence of Class II promoters using CoTrIP, that Rvb1/2 binds specifically at Class II promoters using ChIP-seq, that Rvb1/2 are bound to transcripts with Class II promoters using RIP-Seq, that tethering of Rvb1/2 to a transcript decreases its translatability and that Rvb1/2 binding to a transcript increases its transcript levels by increasing transcription and not slowing mRNA decay.The CoTrIP experiment is clever and for the most part well executed. The key conclusions are largely convincing but some clarifications are nevertheless needed (see below). Overall, this paper is well written with well executed experiments that largely support the authors' model. No major additional experiments are needed to support the claims of the paper. There are a few minor concerns that should be addressed before this manuscript gets published. These are:Minor comments:1) Are Rvb1/2 components (enriched in) of P bodies? The model proposed by the authors suggests this but no data is show.

We find minimal overlap between P-bodies and Rvb1/2 by microscopy – Figure 1—figure supplement 2. We describe this in the text, as well have added reference to previous results also not seeing overlap between cytoplasmic Rvb1/2 granules and P-bodies. “Microscopy revealed that Rvb1/Rvb2 are predominately present in the nucleus when cells are not stressed but a portion of them becomes localized to cytoplasmic granules that are distinct from P-bodies after 30-minute glucose starvation conditions. Similar results were previously seen with 2-deoxyglucose driven glucose starvation, where Rvb1 formed cytoplasmic foci independent of P-bodies and stress granules (Rizzolo et al., 2017).”

2) Figure 1A: The model proposed by the authors indicates that Rvb1/2 and other proteins are recruited to the mRNAs in a promoter-dependent manner and not mRNA sequence dependent manner. This is largely supported by the data presented in the paper. However the authors should also discuss the possibility that RNA sequences could nevertheless contribute as only a uniform ORF has been tested. Could the promoter recruit Rvb1/2 similarly regardless of the ORF sequence tested? Please provide a sequence of the uniform ORF, discuss what this "uniformity" means and how a change in RNA sequence could affect the outcome of the experiment outlined in Figure 1A.

The uniform ORF is CFP, which was also used for the promoter swap experiments in Figure 3B,C. These constructs were originally used in our previous manuscript (Zid and O’Shea 2014) and more detail is given there. We have also added these plasmids to Addgene, where they can be accessed (https://www.addgene.org/178307/). We see similar effects on mRNA localization and protein expression with alternative ORFs such as nanoluciferase used in Figure 4. We have also mixed the 5’UTRs and ORFs of Class I and Class II genes and seen that the promoter is the dominant factor for mRNA localization (Zid lab unpublished data).

3) Figure 2: The authors use Pgk 1 in their ChIP control but this is not the appropriate control for the experiment as Pgk 1 is not nuclear and thus cannot demonstrate non-specific interaction with genetic regions of tested genes. Regardless, the data is convincing enough to support the model that Rvb1/2 are specifically recruited to the promoters of Class II stress-induced genes and not Class I stress-induced genes. GFP-NLS would be a better control. The authors should discuss in their Materials and methods section why they chose a cytoplasmic protein for their normalization control but preferably perform ChIP with GFP-NLS or other nuclear protein that could bind to chromatin non-specifically to further demonstrate the specificity of Rvb1/2 enrichment at Class II promoters.

In the original hyper-ChIPability paper (Teytelman et al. 2015) they found enrichment on highly expressed regions even if there was no-tag present in the strain. We therefore used the highly abundant cytoplasmic protein Pgk1 as a control for pulldown specific enrichment effects, that were unrelated to actual protein binding, and added a note of this in the ChIP-seq methodology section.

4) The authors claim that Rvb1/Rvb2 binding to transcripts leads to formation of granules that are non-colocalized with P-bodies and instead co-localized to SGs, but no SG fluorescent marker is used to demonstrate this claim. The authors should provide this data or remove this claim from their manuscript.

We have removed this claim as we have had issues identifying a robust, endogenously tagged, 30-minute glucose starvation induced stress granule marker and instead refer to these granules as “P-body independent starvation-induced granules”

5) Fluorescent images are fuzzy, very small and difficult to interpret. mRNA puncta are difficult to observe and it is hard to conclude which green puncta colocalize with P bodies and which do not (and how frequently). It is difficult to differentiate between the cytoplasm and nucleus. Consider adding DAPI overlay.

We have added clearer images of mRNA puncta showing that there is minimal overlap between P-bodies and RVB-tethering induced RNA granules (Figure 4C).

6) The relevance of Figure 2B is not clear – please discuss.

From the text “More generally we found that, for genes that show a greater than 3-fold increase in mRNA levels during glucose starvation, their promoters are significantly more enriched for Rvb2 binding. Previously we had found that Hsf1-binding sequences were sufficient to exclude mRNAs from mRNP granules during glucose starvation (Zid & O’Shea, 2014). Interestingly we found that glucose starvation induced Hsf1-target promoters have no difference in Rvb1/Rvb2 binding than an average gene, and significantly lower Rvb1/Rvb2 enrichment than stress induced non-Hsf1 targets (Figure 2B).” This data is also referenced in the discussion “As we found that Rvb1/Rvb2 are generally enriched on the promoters of transcriptionally upregulated mRNAs we favor a model in which the default is for Rvb1/Rvb2 to be recruited to active transcription sites. This fits with previous data that Rvb1/Rvb2 are required to maintain expression of many inducible promoters including galactose-inducible transcripts (Jónsson et al., 2001). While Rvb1/Rvb2 are generally recruited to the promoters of induced mRNAs during glucose starvation, we find that Hsf1-regulated promoters circumvent this recruitment through an unknown mechanism, as the transcriptionally upregulated Hsf1 targets show reduced recruitment relative to non-Hsf1 targets (Figure 2B). Intriguingly, Hsf1-regulated genomic regions have been found to coalesce during stressful conditions (Chowdhary, Kainth, & Gross, 2017; Pincus et al., 2018). It will be interesting to explore whether Rvb1/Rvb2 may be excluded from these coalesced regions in future studies.” Finally, this also serves as a control that Rvb1/Rvb2 are not just binding to transcriptionally induced promoters, as Hsf1 targets are induced just as strongly as Class II mRNAs.

7) Figure 5A modeling adds little supporting evidence to the entire figure. The experimental results are more convincing. Consider moving to the Supplement.

We have moved this to a Figure 4—figure supplement 10A,B.

8) Figure 4 and 3B. The authors suggest that Rvb1/2 loaded by the promoters onto the mRNA determine accumulation of mRNAs to P bodies. To test this model, the authors tether Rvb1/2 onto the mRNA using MS2-MCP system and then look for co-localization of the mRNA with P bodies. However, if the authors' model is correct, this experiment could have been achieved already using the constructs in Figure 3B. The authors should look at the P body localization pattern using chimeras used in Figure 3B.

This experiment was performed in our previous manuscript (Zid and O’Shea 2014) and showed that the promoter was sufficient to determine the cytoplasmic localization of the mRNA but did not show the mechanism for this promoter-controlled localization. One additional clarification is that our data supports Rvb1/2 driving localization to P-body independent starvation-induced granules, which is also the localization pattern we previously saw for Class II promoters.

9) Figure 6: The authors present a model where mRNAs transcribed from Class II promoters are decorated with Rvb1/2 co-transcriptionally, exported into the cytoplasm, recruited to P bodies and translationally repressed. However, this model is not fully supported by the data shown. Specifically, the authors have not shown that localization of mRNAs to P bodies induces translational repression or whether the recruitment is a consequence of this repression. The authors should revise their model to reflect this uncertainty. Also, the numbering of steps 1,2 3 is confusing. Does it imply a temporal sequences? Some of these steps could be occurring simultaneously (like 1 and 3). How does step 3 lead from step 2? Please clarify this model.

We have removed the numbering system as we agree that some of these steps, like 1 and 3 are probably happening at the same time. We have updated the text of the figure legend to “Then Rvb1/Rvb2 escort the interacting mRNAs to the cytoplasm and cause repressed translation and localization to cytoplasmic granules.” We have also updated the discussion with the following text pointing to the unclear causality of translational repression and P-body independent starvation-induced granule localization in the discussion – “We are uncertain whether Rvb1/Rvb2 tethering represses translation which directs mRNAs to mRNP granules, or if Rvb1/Rvb2 binding directly target the mRNA to the granule, which represses translation or some combination of both as these are very hard to disentangle”

10) Consider showing data-points in Figure 1 figure supplement 1. The box/whisker plot doesn't give a good sense of the enrichment alone.

We have added the number of replicates to the figure and have also included the data points in the accompanying supplemental data file for Figure 1 figure supplement 1. They are very close and end up overlapping when plotted on a graph together.

11) Figure 1 Figure supplement 2 shows that the fluorophore seems to influence the % of cells with foci. Why is this the case?

While we can’t be certain, one reason may be that mNeonGreen is ~4 times as bright as mRuby2 in yeast (Botman et al. 2019). This may make it easier to distinguish mNeonGreen Rvb foci versus non-foci fluorescence and the inherent autofluorescence of yeast.

12) List gene names in Figure 2 Figure supp 5.

We have revised Figure 2 Figure supp 5 by adding gene names.

13) Throughout the paper the graph axis labels are very small and difficult to read.

We have increased the font on many graph axis labels.

14) Figure 4 Figure supplement 7C and 8E: on the y-axis the legend says proportion of cells (%), so the value on the y-axis might be 25, 50, 75 and not 0.25, 0.50 and 0.75.

We have fixed these axis values.

15) The last paragraph of the Introduction (page 2) detailed how Rvb1/Rvb2 are core components of the stress granule. Yet most experiments were conducted to relate Rvb1/Rvb2 with P-bodies. Maybe some information about the known roles Rvb1/Rvb2 play in the P-bodies in the Introduction section could help.

We predominantly used a P-body marker because it gives a robust mRNP granule marker during glucose starvation, even at early timepoints. We see minimal colocalization between RVB-tethering induced RNA granules and P-bodies. This is in agreement with ours and others data that RVB doesn’t colocalize with P-bodies during glucose starvation. To our knowledge there is little information linking Rvb1/Rvb2 to P-bodies.

Reviewer #2 (Significance (Required)):Ruvb helicase has been shown to regulate the formation of stress granules in human U2OS cells during oxidative stress (Parker lab, Cell, 2016). Thus, the authors suggest that Rvb proteins could have a broad and conserved role in the formation of RNA granules, which advances our understanding of how biomolecular condensates could form.In addition, translationally-repressed mRNAs have been shown to preferentially recruit to diverse RNA granules, from stress granules P bodies in human cells as well as germ granules in *C. elegans* and *Drosophila*. These observations have gained considerable attention in the past 5 years and exact molecular principles behind this phenomenon are not entirely clear. Long and exposed RNA sequences are thought to be sufficient for this enrichment. The authors however suggest that specific proteins (Rvb1/2) could also trigger enrichment either directly by interacting with P bodies or indirectly by repressing translation and exposing RNA sequences. This finding will be particularly relevant to the field of biomolecular condensates.My expertise is in the area of RNA biology, mRNA decay, RNA granules and mRNA localization.Reviewer #3 (Evidence, reproducibility and clarity (Required)):Dr. Brian Zid has previously published in Nature that, in response to glucose starvation, promoters of some genes ("class II") can control synthesis of mRNAs that are sequestered in cytoplasmic P bodies or Stress granules, away from the translation apparatus. In this paper, his group reports about the underlying mechanism. They have found proteins that bind preferentially class II promoters as well as their transcripts and are capable of repressing their translation and stimulating their assembly with P bodies. They found a correlation between the capacity of Rvb1/2 binding to promoters and binding to mRNAs. Using a tethering technique, they found that Rb1/Rvb2 recruitment to reporter mRNA (not class II) led to the association of the transcript with PBs and its translation repression. Interestingly, Binding of Rvb1/Rvb2 to the studied transcript increased transcription of its own gene, probably by remodeling the nearby chromatin.The paper uncovers a mechanism to sequester mRNAs as translationally repressed in RNA granules during starvation and warrants a publication in a good journal, after responding to various comments below.1. CoTrIP is a method to identify proteins that differentially bind plasmids carrying different promoters/genes. However, the claim that it identifies proteins bound to nascent mRNAs is an overreach, as the proteins bind both DNA and RNA and the purified plasmid contains both types of nucleic acids.Therefore, the title of section 1 ("Rvb1/Rvb2 are identified as potential co-transcriptionally loaded protein factors on the alternative glucose metabolism genes") should be changed to something like: Rvb1/Rvb2 are identified as proteins that are co-purified with a plasmid expressing alternative glucose metabolism genes.Description of CoTrIP and its results should be discussed throughout the manuscript accordingly.

We changed the title of the section to: “Rvb1/Rvb2 co-purify with plasmids containing an alternative glucose metabolism gene promoter”

2. The engineered Rvb1/Rvb2 tethering to mRNAs of choice is a potentially convincing way to show the causative effect of Rvb1/Rvb2 on RNA performance. Using this method, the authors show that attachment of Rvb1/Rvb2 to an engineered mRNA mediate its association with granules and inhibits its translation. However, this experiment takes Rvb1/2 out of its natural context such that its behavior in this case may not be exemplative of its endogenous function. The authors are encouraged to support their results by depleting Rvbs with AID and examine the outcome of this depletion on PBs formation and translation of class II genes (and class I as controls).

We had extensively tried to use the AID system to deplete Rvbs, but had leaky depletion even without auxin addition with all AID systems we tried. We alternatively used CRISPRi and a gRNA directed to the promoter of RVB2 to downregulate expression of RVB2. This led to an almost 20-fold depletion of the RVB2 mRNA. From this strain we found increased protein induction for Class II genes during glucose starvation, but no effect on a Class I control. Even though the Class II genes had increased protein expression, we actually found decreased mRNA induction upon downregulated RVB2 expression during glucose starvation, further supporting the role of Rvb proteins affect transcription and translation. This data is presented as Figure 5.

3. The tethering experiments, shown in Figure 4, would be more convincing by including an additional control. To rule out the possibility that any bulky protein that is recruited to the 3'-UTR by the PP7 element affects translation (not an unlikely possibility), they want to consider fusing irrelevant protein (e.g., Pgk1p) to CP, in place of Rvb1/2.

We have expressed GFP-PP7-CP plasmid in the HSP30prUTR-nLuc-PP7-MS2 yeast strain and found that this tethering has no significant effect on protein synthesis induction, mRNA induction, and translatability (Figure 4B).

4. The proposal that Rvb1 binds class II transcripts during transcription is a plausible possibility (which I personally believe to represent the reality), but by no means demonstrated. This should be clearly addressed in the paper.

We agree with the reviewer that this has not been explicitly shown and have tried to use appropriate language to convey this point that the data suggests and it is likely but not absolutely certain.

Results: “This suggests that only the promoter itself can determine the transcribed mRNA’s interaction with Rvb1/Rvb2, further indicating that Rvb1/Rvb2 are likely to be co-transcriptionally loaded from the promoters to nascent mRNAs.”

Discussion: “suggesting Rvb1/Rvb2 are loaded from enriched promoters to the nascent mRNAs.”

5. An optional suggestion: The paper can be upgraded by performing ribosome profiling, as shown in Supplemental Figure 9, after a short depletion of Rvb1/2 by AID (see comment 2). This, in combination with the results already shown in Supp Figure 9, can demonstrate the role of Rvb1/2 in mRNA storage in granules and in translation shortly after glucose refeeding. The large data sets thus produced (in particular the ratio between depleted and non-depleted signal per each gene) can be used to try correlate the extent of ribosome occupancy (or the above mentioned ratio) with cis-element(s) or known trans-acting elements within the promoters. This may identify elements within the promoters that recruit (directly or indirectly) Rvb1/2. If successful, it can pave the way to demonstrate co-transcriptional RNA binding. I also suggest moving Supp Figure 9 as an additional panel of the main Figure 4.

We agree that it would be an interesting direction to pursue ribosome profiling on the RVB2 depletion strain. Currently due to time constraints and graduating students we have not been able to perform this experiment.

Minor point:1. The original reference about "mRNA imprinting" was published by Choder in Cellular logistics 2011.

We have added this reference to the manuscript.

2. The graph in 5B does not have error bars and the number of replicates is unclear.

We have added the number of biological replicates to the figure legend (n=4) and statistical analysis to Figure 5B (now Figure 4E).

Reviewer #3 (Significance (Required)):The paper uncovers a mechanism to sequester mRNAs as translationally repressed in RNA granules during starvation. This significantly advances our understanding of how gene expression in yeast responds to the environment and warrants a publication in a good journal, after responding to the various comments, indicated above.My expertise is regulation of gene expression.Referee Cross-commentingIn general all reviewers feel that the paper deals with a significant issue, each from his/her point of view, and is basically of high quality.I concur with all the comments of Reviewer 1 and 2. In particular, two comments drove my attention.Reviewer 1: Would be important to see increase in translation after glucose replenishment in one of the reporter constructs used in the paper, such as GL3 promoter driven CFP.

We have tagged Class II mRNAs (GSY1 and HXK1) as well as Class I mRNAs (HSP30 and HSP26) and found that the Class II reporters show more robust induction upon glucose re-addition than Class I mRNAs further supporting our ribosome profiling data (Figure 4—figure supplement 9B).

Reviewer 2: The authors should look at the P body localization pattern using chimeras used in Figure 3B.

This experiment was performed in our previous manuscript (Zid and O’Shea 2014) and showed that the promoter was sufficient to determine the cytoplasmic localization of the mRNA as the HSP26 promoter drove diffuse localization, while the GLC3 promoter led to RNA granule formation, that was mostly distinct from P-bodies, but sometimes overlapping. In the previous manuscript we were not able to show the mechanism for this promoter-controlled localization.